



# Oblique reactivation of lithosphere-scale lineaments controls rift physiography – The upper crustal expression of the Sorgenfrei-Tornquist Zone, offshore southern Norway

Thomas B. Phillips [1], Christopher A-L. Jackson [1], Rebecca E. Bell [1], Oliver B. Duffy [2]

[1] Basins Research Group (BRG), Department of Earth Science and Engineering, Imperial College, South Kensington Campus, Prince Consort Road, London, SW7 2BP, UK

[2] Bureau of Economic Geology, Jackson School of Geosciences, The University of Texas at Austin, University Station, Box X, Austin, TX 78713-8924, USA

*Correspondence to*: Thomas B. Phillips (t.phillips13@imperial.ac.uk)

**Abstract.** Pre-existing structures within sub-crustal lithosphere may localise stresses during subsequent tectonic events, resulting in complex fault systems at upper crustal levels. As these sub-crustal structures are difficult to resolve at great depths, the evolution of kinematically and perhaps geometrically linked upper-crustal fault populations can offer insights into their deformation history, including when and how they reactivate and

accommodate stresses during later tectonic events. In this study, we use borehole-constrained 2D and 3D seismic reflection data to investigate the structural development of the Farsund Basin, offshore southern Norway; this E-trending basin represents the upper crustal expression of the Sorgenfrei-Tornquist Zone, a major lithosphere-scale lineament extending >1000 km across Central Europe. The southern margin of the Farsund Basin is characterised by N-S and E-W-striking fault populations, the latter extending down through the Moho

and potentially linking with the Sorgenfrei-Tornquist Zone as imaged within sub-crustal lithosphere. Due to this geometric linkage, we can analyse the upper crustal fault populations to infer the kinematics of the Sorgenfrei-Tornquist Zone. We use throw-length (T-x) analysis and fault displacement backstripping techniques to determine the geometric and kinematic evolution of upper-crustal fault populations during the multiphase evolution of the Farsund Basin. We document a period of sinistral strike-slip activity along E-W-striking faults

during the Early Jurassic, representing a hitherto undocumented phase of activity along the Sorgenfrei-Tornquist Zone. These E-W-striking upper-crustal faults are later obliquely reactivated under a dextral stress regime during the Early Cretaceous, with new faults also propagating away from pre-existing ones, representing a switch to a phase of dextral transtension along the Sorgenfrei-Tornquist Zone. We show that the Sorgenfrei-Tornquist Zone represents a long-lived lithosphere-scale lineament that is periodically reactivated throughout its

protracted geological history. The upper crustal component of the lineament is reactivated in a range of tectonic styles, including both sinistral and dextral strike-slip motions, with the geometry and kinematics of these faults often inconsistent with what may otherwise be inferred from regional tectonics alone. Understanding these different styles of reactivation not only allows us to better understand the influence of sub-crustal lithospheric structure on rifting, but also offers insights into the prevailing stress field during regional tectonic events.



## 1 Introduction

Pre-existing structures, such as prior fault populations, shear zones and terrane suture zones, are present throughout the lithosphere, where they may influence the geometry and evolution of upper crustal rift systems forming during later tectonic events (e.g. Daly et al., 1989; Mogensen, 1995; Doré et al., 1997; Morley et al.,

2004; Graversen, 2009; Gontijo-Pascutti et al., 2010; Bellahsen et al., 2013; Bird et al., 2014; Whipp et al., 2014; Bladon et al., 2015; Salomon et al., 2015; Phillips et al., 2016; Brune et al., 2017). The geometry and origin of pre-existing structures in the upper crust, such as pre-existing faults or shear zones, can often be directly imaged, i.e. in the field or on seismic reflection data, thus allowing their role during subsequent rifting to be investigated (e.g. Kirkpatrick et al., 2013; Reeve et al., 2013; Bladon et al., 2015; Phillips et al., 2016;

Fazlikhani et al., 2017). However, the geometry and physical properties of deeper-lying structures in sub-crustal lithosphere are less well constrained, with information provided primarily by whole crust to lithosphere imaging geophysical methods such as seismic tomography, deep seismic reflection surveys, seismic refraction surveys, and potential field imaging. Although able to image these structures to substantial, sub-crustal depths, such techniques are relatively low resolution, thereby limiting our ability to interpret the geological origin of such

structures, and thus hampering efforts to examine how they may influence the structural style and kinematics of later formed rift systems.

In previously rifted areas, structures within the sub-crustal lithosphere are often associated with complex upper crustal rift systems, which may locally follow structural trends oblique to those predicted by extension of homogeneous lithosphere (Holdsworth et al., 2001; Tommasi and Vauchez, 2001; Bergerat et al., 2007;

Graversen, 2009; Le Breton et al., 2013; Daly et al., 2014). Although exactly how these anomalous rift systems link to deeper structures is uncertain, their geometry and kinematic evolution can record the regional tectonic history, often throughout multiple stages of reactivation and under the influence of pre-existing structures at deeper levels (e.g. Mogensen, 1994; Bergerat et al., 2007; Corti, 2009; Brune et al., 2017). If we are able to establish a link between these sub-crustal structures and upper crustal fault populations, we can use the

evolution of the latter to examine the kinematic response of structures in the sub-crustal lithosphere during regional tectonic events. To accomplish this we first need data that allows us to examine and link the structures at both deep and shallow levels. We can then use high-resolution datasets, focussed at shallower levels, to extract detailed information which can be applied to the evolution of the whole system.

In this study we use borehole-constrained 2D and 3D seismic reflection data to analyse the geometric and

kinematic evolution of an upper crustal fault population on the southern margin of the E-trending Farsund Basin, offshore southern Norway (Fig. 1a). The Farsund Basin is situated above the NW- to W-trending Sorgenfrei-Tornquist Zone (STZ), a major pan-Central European lineament that is defined by a sharp change in lithospheric thickness at sub-crustal depths (Wylegalla et al., 1999; Cotte and Pedersen, 2002; Babuška and Plomerová, 2004; Hossein Shomali et al., 2006; Mazur et al., 2015). The Farsund Basin is characterised by E-W- and N-S-

striking upper crustal fault sets that have been periodically active throughout the multiphase tectonic evolution of the North Sea. We first establish a geometric link between the seismically imaged upper crustal faults defining the Farsund Basin, and the change in lithospheric thickness at depth, arguing that the former is geometrically linked to and represents the upper crustal expression of the latter. Having established this





geometric link, we then analyse the detailed geometric and kinematic evolution of the upper crustal fault sets,

using the observed deformation history to infer the kinematic response of the lithosphere-scale STZ to several regional tectonic events. We show that N-S striking faults were active during the Triassic, with some apparent activity occurring along segments of E-W faults at this time. The main period of activity along the main, E-W-striking faults occurred later, during the Early Cretaceous. Our quantitative fault analyses highlight a previously unrecognised period of sinistral strike-slip activity along E-W-striking faults during the Early-Middle Jurassic.

During Late Jurassic-Early Cretaceous extension, the stress regime switches, with dextral transtensional activity occurring along the E-W-striking faults.

The E-W-striking faults defining the Farsund Basin represent the upper crustal component of the lithosphere-scale STZ. These faults, which are often non-optimally oriented with respect to the regional stress field, are reactivated in a range of styles during multiple tectonic events, representing activity along the whole

lithosphere-scale STZ. We find that the sense of motion and style of reactivation along the STZ, as identified from the upper-crustal fault populations, reflects the prevailing regional stress field during these tectonic events.

## 2 Geological setting

This study examines a c. 1000 km$^2$ area offshore southern Norway, focussing on the western Farsund Basin (Fig. 1). The E-trending basin is bordered to the south by the Norwegian Danish Basin and to the north by the N-

trending Eigerøy Horst, Varnes Graben and Agder Horst, from west to east respectively (Fig. 1c). The basin extends westwards to the Stavanger Platform and merges with the Danish sector of the Norwegian-Danish Basin to the east (Liboriussen et al., 1987; Christensen and Korstgård, 1994) (Fig. 1a). The basin overlies the inferred westernmost extent of the STZ (Pegrum, 1984) (Fig. 1a).

### 2.1 Geometry and origin of the Sorgenfrei-Tornquist zone

The STZ represents the northwestern segment of the Tornquist Zone, a lineament that spans the lithosphere and extends >1000 km across Central and Northern Europe (Pegrum, 1984; Berthelsen, 1998; Mazur et al., 2015; Alasonati Tašárová et al., 2016). The Tornquist Zone comprises two segments: the Teisserye-Tornquist zone (TTZ) in the south, extending northwest from the Carpathian orogenic front to the Rønne Graben (e.g. Guterch et al., 1986; Berthelsen, 1998; Grad et al., 1999; Pharaoh, 1999; Alasonati Tašárová et al., 2016) (Fig. 1a), and

the STZ in the north, continuing northwest from the Rønne Graben to the Farsund Basin (e.g. Pegrum, 1984; Berthelsen, 1998; Thybo, 2000; Babuška and Plomerová, 2004), and possibly extending further westwards beneath the main North Sea rift (Pegrum, 1984) (Fig. 1a).

The Tornquist Zone, including the STZ and TTZ, has been extensively studied using a variety of geological and geophysical methods, including seismic tomography (Cotte and Pedersen, 2002; Voss et al., 2006), seismic

refraction (Guterch et al., 1986; Guterch and Grad, 2006; Alasonati Tašárová et al., 2016), seismic anisotropy (Babuška and Plomerová, 2004), and seismic reflection surveying (Grad et al., 1999; Thybo, 2000; Lassen and Thybo, 2012). These data suggest that the lineament separates thick, old cratonic lithosphere of the East European Craton, including Baltica, from the younger, thinner lithosphere associated with assorted Palaeozoic terranes belonging to Central and Western Europe (Fig. 1d); the lineament thus represents a major change in



lithospheric properties and thickness (e.g. Pegrum, 1984; Kinck et al., 1993; Michelsen and Nielsen, 1993; Erlström et al., 1997; Berthelsen, 1998; Pharaoh, 1999; Cotte and Pedersen, 2002; Babuška and Plomerová, 2004; Voss et al., 2006).

At the junction of the TTZ and STZ, offshore southern Sweden, a zone of NW-diverging splay faults, termed the 'Tornquist Fan' occur, demarcated to the north and south by the STZ and Trans-European Fault respectively

(Fig. 1a) (Thybo, 2000, 2001). Here, the STZ is still defined as a change in lithospheric thickness at sub-crustal, i.e. upper mantle, levels (e.g. Berthelsen, 1998; Cotte and Pedersen, 2002; Babuška and Plomerová, 2004); however, at upper crustal levels it is defined by a zone of Late Cretaceous inversion (e.g. Pegrum, 1984; Michelsen and Nielsen, 1993; Mogensen and Jensen, 1994; Deeks and Thomas, 1995; Mogensen, 1995; Bergerat et al., 2007). Geological evidence, in the form of drilled crystalline basement, shows that basement of

Baltica affinity is present either side of the STZ (Berthelsen, 1998); indicating that the margin of Baltica as identified at sub-crustal depths does not correspond to the same location at upper crustal levels, with upper crustal crystalline basement of Baltica affinity continuing south of the STZ (Fig. 1d).

The STZ, as defined at upper crustal levels by the zone of Late Cretaceous inversion, correlates in plan-view to the STZ as defined by the change in lithospheric thickness at sub-crustal levels (Liboriussen et al., 1987;

Mogensen and Jensen, 1994; Babuška and Plomerová, 2004). Several authors suggest the STZ acts as a weak zone during later tectonic events, acting to accommodate stresses between adjacent crustal blocks (Mogensen and Jensen, 1994; Mogensen and Korstgård, 2003); however, the link between different structural levels and, therefore, how a change in lithospheric thickness may behave kinematically and influence the development of overlying upper crustal rift systems later events, has not been established. Understanding how these sub-crustal

structures are expressed within rift systems is vital to understanding both the local rift evolution and the causal stress field.

## 2.2 Regional geological history

Rifts above the STZ have been active during multiple tectonic events, since at least the Carboniferous-Permian

(Jensen and Schmidt, 1993; Michelsen and Nielsen, 1993; Deeks and Thomas, 1995; Mogensen, 1995; Erlström et al., 1997; Thybo, 2000; Mogensen and Korstgård, 2003; Graversen, 2009). Variscan orogenesis drove regional N-S-directed shortening in the Carboniferous-to-Permian, resulting in the formation of a W- to NW-trending strike-slip system, incorporating the W-trending Fjerritslev Faults (Skjerven et al., 1983), in addition to several N-to NE-trending structures such as the Varnes and Skagerrak grabens, situated north and east of the

study area respectively (Fig. 1a) (Ro et al., 1990; Heeremans and Faleide, 2004; Heeremans et al., 2004; Lassen and Thybo, 2012).

These fault systems were intermittently active during the Mesozoic in response to several tectonic events affecting the broader North Sea (e.g. Liboriussen et al., 1987; Mogensen and Jensen, 1994; Deeks and Thomas, 1995; Mogensen, 1995; Erlström et al., 1997; Mogensen and Korstgård, 2003; Bergerat et al., 2007). During the

Permian-Triassic, E-W continental extension occurred in response to the breakup of Pangea, leading to the formation of a predominately N-trending rift across the North Sea (e.g. Ziegler, 1992; Færseth, 1996; Bell et al., 2014). In the Horn Graben, south of the Farsund Basin, rifting and associated activity on N-S striking faults

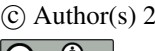



initiated during the Late Permian (Vejbæk, 1990) (Fig. 1a). Rifting migrated northwards into the Norwegian-Danish Basin during the Triassic, towards the southern margin of the study area, where additional N-S-striking

faults occur (Fig. 1c). An additional rift phase, centred on the Central North Sea and related to the collapse of a Middle Jurassic thermal dome (Rattey and Hayward, 1993; Underhill and Partington, 1993), initiated in the Late Jurassic, continuing until the Early Cretaceous. The extension direction during this rift phase varies spatially, with NW-SE to E-W extension proposed for areas north of the study area in the main northern North Sea (e.g. Brun and Tron, 1993; Færseth, 1996; Doré et al., 1997; Bell et al., 2014), and NE-SW extension proposed south of the

study area in the Central Graben and southern North Sea (Coward et al., 2003). During the Late Cretaceous, horizontal shortening, related to far-field tectonic effects from the Alpine Orogeny, drove basin inversion within several North Sea sedimentary basins (Biddle and Rudolph, 1988; Cartwright, 1989; Mogensen and Jensen, 1994; Jackson et al., 2013; Phillips et al., 2016). This inversion was amplified along the upper-crustal expression of the STZ (Thybo, 2000), being associated with regional uplift and the reverse reactivation of basin-bounding faults

(e.g. Liboriussen et al., 1987; Jensen and Schmidt, 1993; Mogensen and Jensen, 1994; Deeks and Thomas, 1995; Bergerat et al., 2007). Further uplift and erosion occurred during the Neogene due to uplift of the South Swedish Dome (Jensen and Schmidt, 1993; Japsen et al., 2002), resulting in erosion of Cretaceous strata across much of the study area.

The E-trending Farsund Basin is oriented at an unusually high angle, in many cases almost perpendicular, to largely N-trending structures across the North Sea. To the south, in the Horn and Central grabens, faults primarily strike N-S to NW-SE (Vejbæk, 1990; Glennie, 1997), whereas to the north and west, where they define the Varnes Graben and Lista Nose, they strike N-S (Skjerven et al., 1983; Heeremans et al., 2004; Lewis et al., 2013) (Fig. 1a). Furthermore, to the east, NE-SW and NW-SE striking faults occur in the Skagerrak Graben and along the eastern part of the STZ respectively (Ro et al., 1990; Mogensen and Jensen, 1994) (Fig. 1a). The E-W geometry

of the faults defining the Farsund Basin mean that they are not optimal for reactivation during any of the North Sea regional tectonic events outlined above, suggesting they record a hitherto undocumented phase of tectonic activity or, more likely, are influenced by a pre-existing structure such as the STZ.

### 3 Data and methods

#### 3.1 Data

Seismic interpretation focused on a 500 km$^2$, borehole-constrained 3D seismic reflection dataset covering the southern margin of the Farsund Basin (Fig. 1c). These data image to 4 s two-way-time (TWT) (c. 6 km), having inline and crossline spacings of 18.75 m and 12.5 m respectively. A regional 2D seismic reflection dataset covering the entire basin was used to provide regional structural context for the area imaged by the 3D dataset; this 2D dataset consists of closely-spaced (c. 3 km), N-trending seismic sections that are oriented perpendicular

to the dominant E-W structural trend (see Fig. 1c for the locations of sections used in this study). Seismic reflection data are zero-phase and follow the SEG normal polarity convention whereby a downward increase in acoustic impedance is represented by a peak (black) and a downward decrease in acoustic impedance is represented by a trough (red). Image quality is excellent throughout the 2D and 3D datasets at shallow levels, although quality deteriorates at depth; the 2D sections image to deeper structural levels (7 s TWT), with some reaching 12 s TWT,

and cover a wider area (c. 2000 km$^2$) than the 3D dataset. The ages of key horizons are constrained using



stratigraphic information from well 11/5-1, located in the area covered by the 3D dataset, and wells 9/3-1, 10/5-1, 10/7-1, 10/8-1 and 11/9-1, located in the wider region imaged by the 2D seismic data (Fig. 1a). Checkshot information from these wells was used to convert structural measurements and thickness from the time to the depth domain. We mapped seven key tectono-stratigraphic horizons and generated isochrons between them to

define the structural style infer the subsidence history of the basin. Cut-offs along these horizons provided the inputs to the quantitative fault analysis.

### 3.2 Quantitative fault analysis

To constrain the geometry and growth of the upper-crustal fault population, we mapped and performed

quantitative analysis of the major faults delineating the basin. Horizon cut-offs, fault tip-lines and fault intersections (as defined by branchlines) were mapped to minimise artefacts that may lead to an incorrect assessment of fault kinematics (e.g. Walsh et al., 2003; Duffy et al., 2015; Yielding, 2016). Throw-length (T-x) plots, where throw is measured at regular intervals along the fault for different stratigraphic levels (see Appendix A), were then calculated, and were subsequently used as input for fault-displacement backstripping, a

technique used to unravel the kinematic and geometric evolution of a fault throughout its history (see Appendix B) (see Jackson et al., 2017). T-x plots allow us to examine the distribution of throw along a fault and help elucidate its kinematic history. Fault displacement backstripping expands upon this; by systematically removing throw accrued during specific stratigraphic intervals, starting with the youngest, we are able to quantitatively examine the throw distribution along the fault throughout its history, and hence determine its geometric

evolution.

### 4 Structural style of the Farsund Basin

In Sect. 4.2, we use seismic sections and two-way-time (TWT) structure maps to outline and describe the upper crustal fault system defining the Farsund Basin, establishing a structural framework for use in later sections. First, in Sect. 4.1, we document the crustal-scale fault geometries associated with the Farsund Basin and attempt

to, at least geometrically, link them to the sub-crustal lithospheric 'step' defined as the STZ.

### 4.1 Crustal-scale faulting

The E-W-striking Fjerritslev North and Fjerritslev South Faults delineate the southern margin of the Farsund Basin; these faults merge laterally to the east to form a single structure, the Fjerritslev Fault system (Fig. 1c). A S-dipping fault, termed the Farsund North Fault, bounds the Farsund Basin to the north (Fig. 2), separating it

from N-S striking faults associated with the Varnes Graben (Fig. 1c). Projecting these basin-bounding faults downwards, based on their overall dip and subtle reflection terminations at depth, the N-dipping Fjerritslev North and South faults appear to merge at a depth of c. 7 s TWT (15 km) with the S-dipping Farsund North Fault, as well as several other faults within the Varnes Graben (Fig. 2). Together, these structures thus appear to form a system that, at least superficially, resembles a negative flower structure (Cheng et al., 2017) (Fig. 2). The

Fjerritslev North and South faults also merge eastwards (Fig. 1c). Although the exact location of the apparent geometric linkage at depth is uncertain due to poor seismic image quality, we infer that the faults defining the basin are geometrically linked, both along-strike and down-dip (Fig. 2).



We observe high-amplitude reflections at c. 9-10 s TWT at the basin margins; we interpret these as Moho-related reflections, which are noticeably absent directly beneath the Farsund Basin (Fig. 2). In addition, east of the Farsund Basin, Moho-related reflectivity is observed at 10-12 s TWT (c. 30 km); a key observation in this location is that this reflectivity is offset by 1-2 s TWT (4-5 km) across the Fjerritslev Fault System (Lie and Husebye, 1994; Kind et al., 1997). Based on these observations, we propose faults defining the Farsund Basin offset the Moho, causing it to be poorly imaged below the basin. Furthermore, we speculate that crustal-scale faults extend deeper below the Moho to link to the lithospheric step associated with the STZ as defined at sub-crustal levels (Fig. 1c, 2) (e.g. Berthelsen, 1998; Cotte and Pedersen, 2002; Babuška and Plomerová, 2004). Based on the anomalous, overall E-W strike of the basin-bounding faults compared to the broader northerly trending North Sea rift (Fig. 1a) and the whole-crust spanning geometry of the fault system (Fig. 1c, 2), we propose that faults defining the Farsund Basin represent the upper crustal expression of the STZ. We therefore argue that, by studying the geometric and kinematic evolution of these well-imaged upper-crustal fault systems, we can gain a better understanding of the kinematic behaviour of the whole lithosphere-scale STZ.

### 4.2 Upper-crustal fault geometries

Where present, the base of the Upper Permian Zechstein Supergroup salt represents the deepest mappable coherent reflection within the study area (i.e. top Acoustic Basement; Fig. 3, 4). Where Zechstein salt is absent and Triassic strata directly overlie pre-Upper Permian strata, or where erosion by the Base Jurassic Unconformity (BJU) removes Triassic strata completely, we map these basal reflections (i.e. Base Triassic or BJU) as the Acoustic Basement (Fig. 4, 5). In plan-view, the Acoustic Basement surface is characterised by the E-W-striking Fjerritslev North and South faults and a N-trending fault population (Fig. 3, 4). The Fjerritslev North Fault is c. 70 km long (32 km within the 3D volume) (Fig. 1a, 3), with a series of relay ramps separating the fault into three, c. 10 km long segments within the area covered by the 3D volume (Fig. 3). The Fjerritslev South Fault is c. 75 km long (38 km within the 3D volume) (Fig. 1a), and has several branchlines with N-S-striking faults (Fig. 3).

Two major E-dipping, N-S-striking faults, hereby termed NS1 and NS2 from north to south respectively, dissect the basin (Fig. 3, 5). NS1 and NS2 lie in the hanging wall and footwall of the Fjerritslev North Fault respectively and also abut this structure. The two branchlines are laterally offset by c. 10 km. Further south, NS2 cross-cuts the Fjerritslev South Fault (Fig. 3). We observe additional N-striking, E-dipping faults within the hanging walls of NS1 (i.e. HF1) and NS2 (i.e. HF2; Fig. 5). HF2 links and terminates against the Fjerritslev North and South faults (Fig. 3). Another N-S striking fault, termed NS3, is located east of HF2 where it abuts the footwall of the Fjerritslev South Fault (Fig. 3). A series of minor (c. 50 ms TWT throw), N-S striking faults are present across the footwall of NS2; these faults cross-cut the Fjerritslev South Fault and are eroded by the BJU to the north, terminating at the Acoustic Basement surface within the footwall of the Fjerritslev North Fault (TF1-4; Fig. 3, 5). 2D seismic data indicate that, south of the 3D dataset, these faults may merge into a single structure (Fig. 1c).

Supra-salt structural levels (i.e. BJU and above) are dominated by the E-W Fjerritslev North and South faults (Fig. 4), with a key observation being that N-S striking faults, with the exception of NS2, are absent (Fig. 3, 5). At these structural levels the Fjerritslev South Fault terminates at the NW-SE striking HF2 (Fig. 3). Numerous





faults, displaying a range of strikes, are present within the footwall of the Fjerritslev South Fault at supra-salt
structural levels; these represent thin-skinned, salt-detached faults that are accordingly not expressed at subsalt
structural levels (i.e. the Acoustic Basement) (Fig. 2, 4). No thick-skinned (i.e. basement-involved) faults are
present at the top of the Lower Cretaceous, the only faults present being arcuate, broadly E-W oriented, S-
dipping, salt-detached faults located along the footwall of the Fjerritslev South Fault (Fig. 3, 4).

**5 Tectono-stratigraphic evolution of the Farsund Basin**

Having established the present structural style of the Farsund Basin, we here integrate observations from N- and
E-trending seismic cross-sections (Fig. 4, 5) and sediment isochrons (Fig. 3, 6) to broadly constrain the spatio-
temporal patterns of faulting during basin development.

**5.1 Pre-upper Permian**

Regional 2D seismic reflection data indicate the acoustic basement is reflective, but contains very little in the
way of coherent, mappable reflectivity. Discrete reflections likely represent Carboniferous-Permian strata (Fig.
4, 5) (e.g. Mogensen and Jensen, 1994; Sørensen and Tangen, 1995). Acoustic basement reflections are
truncated at base salt (or its laterally equivalent stratigraphic horizon where salt is absent) (Fig. 4).
Carboniferous-Permian strata are tabular in both the hanging walls and footwalls of the Fjerritslev North and
South faults, indicating these structures were inactive. Apparent thickening of Late Palaeozoic strata into the
hanging wall of the Fjerritslev North Fault appears to simply reflect increased updip erosion of strata below the
BJU (Fig. 4). Syn-kinematic strata also appear absent in the hanging wall of NS2, although this may be due to
poor at-depth imaging within the 3D volume (Fig. 5). Instead, Carboniferous-Permian strata appear to thicken
regionally to the south (Fig. 2, 4). The seismic-stratigraphic architecture of sub-salt strata suggests E-W-striking
faults were inactive during the Carboniferous-to-Permian (Fig. 4, 5).

**5.2 Triassic**

The Triassic interval does not thicken into the Fjerritslev South Fault along most of its length (Fig. 2, 4),
although a depocentre, which may be related to salt withdrawal and not fault movement, may potentially be
present within the hanging wall of the eastern segment (Fig. 6a). The Triassic interval also does not thicken
towards the eastern part of Fjerritslev North Fault (i.e. east of its branchline with NS2), suggesting at least this
part of the fault was inactive at this time (Fig. 6a). A critical observation we make is that a thick, tabular,
seemingly pre-kinematic package of Triassic strata is preserved within the hanging wall of the central segment
of the E-W-striking Fjerritslev North Fault (i.e. between its branchlines with NS1 and NS2; Fig. 4, 6a). With
regards to activity along the N-S-striking faults, the Triassic interval thickens across NS1 and NS2 (Fig. 5, 6),
with Triassic strata also preferentially preserved in the hanging walls of other N-S faults beneath the BJU (Fig.
6a). Triassic strata are largely eroded across the footwall of NS2 and are completely eroded from the footwall of
NS1, with the thickness of missing section increasing northwards (Fig. 4, 6a). Our observation that Triassic
thickness changes principally reflect differential preservation of strata beneath the BJU indicates that any
activity along depocentre-bounding faults must have occurred prior to BJU erosion in the Early Jurassic (Fig. 5,
6a). Therefore, Triassic activity predominantly occurred on N-S striking faults, with the central segment of the



E-W Fjerritslev North Fault and potentially the eastern segment of the Fjerritslev South Fault also appearing active (Fig. 6a). The other segments of the E-W-striking Fjerritslev North and South faults were inactive at this time (Fig. 2, 4, 6a).

### 5.3 Jurassic

The Jurassic interval is relatively thin across the basin (c. 200 ms TWT, c. 250 m) and of constant thickness within individual fault blocks (Fig. 2, 6b). A subtle, stepwise basinward thickening (c. 60 ms TWT, 75 m) occurs northwards across the Fjerritslev North and South faults (Fig. 6b), with relatively minor thickness changes (c. 40 ms TWT, 50 m) also occurring across NS1 and NS2 (Fig. 4, 6b). Based on the lack of obviously fault-driven, short-wavelength changes in Jurassic sediment thickness, we propose the Middle and Late Jurassic (i.e. post BJU-erosion) were a time of relative tectonic quiescence (Fig. 4, 6b).

### 5.4 Cretaceous

The Lower Cretaceous interval thickens northwards across the E-W-striking Fjerritslev South and North faults, reaching a maximum thickness of c. 1400 ms TWT (1800 m) in the centre of the basin (Fig. 4, 6c). Fault-related thickening is observed along the entire length of the Fjerritslev North Fault, with strata thickening eastwards towards the largest depocentre occurring next to, the eastern segment (Fig. 6c). Three depocentres occur along the Fjerritslev South Fault; a minor, E-trending depocentre in the west (c. 700 ms TWT, 850 m), a major E-trending depocentre situated between the TF1 branchline location and the NS2 branchline (c. 850 ms TWT, c. 1000 m) (Fig. 6c), and a further, NE-trending depocentre to the east of the NS2 branchline (c. 800 ms TWT, c. 900 m) (Fig. 6c). Lower Cretaceous strata do not appreciably thicken across N-S faults, and this interval is partially eroded to the northwest by the Base Cenozoic unconformity (Fig. 4, 6c). Upper Cretaceous strata are largely absent across the basin due to erosion along the Base Cenozoic Unconformity; strata of this age are only preserved along the southern basin margin (Fig. 2, 4).

### 5.5 Summary of basin evolution

Palaeozoic-to-Mesozoic deformation in the Farsund Basin was accommodated by activity on both E-W and N-S-striking faults, although slip was strongly partitioned in time and space. Triassic faulting primarily occurred along N-S striking faults (Fig. 5, 6a), along with isolated segments of E-W-striking faults (Fig. 4, 6a). In contrast, in the Early Cretaceous, following Early-to-Middle Jurassic regional uplift and erosion and a period of relative tectonic quiescence during the Middle and Late Jurassic, activity preferentially occurred on E-W-striking Fjerritslev Faults (Fig. 2, 4, 6c).

## 6 Geometric and kinematic evolution of upper crustal faults

Having constrained, to the first order, the tectono-stratigraphic evolution of the Farsund Basin (Fig. 6), we now analyse the geometric and kinematic evolution of the upper crustal fault populations. To achieve this, fault-displacement backstripping was undertaken on the E-W-striking Fjerritslev North and Fjerritslev South faults (Fig. 7, 8). Additional throw-length plots were generated for the N-S-striking faults along the Acoustic Basement horizon; the faults are largely truncated by the BJU and display negligible throw at shallower, and



therefore younger, stratigraphic levels (Fig. 3). Because the E-W faults tip out within the Lower Cretaceous succession, we use an intra-Lower Cretaceous (ILC) surface to help constrain the geometric and kinematic evolution of the faults during this time interval (Fig. 4, 5).

### 6.1 Triassic fault activity

Triassic extension was concentrated on N-S striking faults, as well as relatively short, discrete segments of the E-W-striking Fjerritslev North and South faults (Fig. 6a). The central segment of the Fjerritslev North Fault, between its branchlines with NS1 and NS2, corresponds to a marked increase in throw (c. 1000 ms TWT, 1300 m) as measured along the Acoustic Basement horizon (Fig. 7a). Throw increases sharply at the branchlines with NS1 and NS2, with significantly larger throws (c. 1500 ms TWT, 2000 m) observed along the central segment compared to the western and eastern segments (c. 500 ms TWT, 650 m) (Fig. 7a). Throw on NS1 is c. 950 ms TWT (1700 m) at the branchline with the Fjerritslev North Fault, decreasing northwards to c. 500 ms TWT (c. 800 m) (Fig. 7a); throw on NS2 at its branchline with the Fjerritslev North Fault is c. 1200 ms TWT (1700 m), decreasing southwards to c. 600 ms TWT (850 m) (Fig. 7a). NS1 and NS2 are eroded along the BJU, therefore throw on these faults as calculated across the Acoustic Basement horizon largely represents throw accrued during the Triassic. NS2 shows relatively minor, post-Triassic activity (Fig. 5, 9)

Due to BJU erosion and the related absence of Triassic strata, the western segment of the Fjerritslev North Fault only records post-BJU activity (Fig. 6a). Along the eastern segment of the fault, where Triassic strata are preserved, the same amount of throw occurs as along the western segment (c. 500 ms TWT, 650 m), indicating that both segments were only active post-BJU, likely during the Early Cretaceous (Fig. 7). Only the central segment of the Fjerritslev North Fault shows any pre-Early Cretaceous activity, with throw backstripping showing that the central segment of the Fjerritslev North Fault, along with NS1 and NS2, accrued c. 1000 ms TWT (1300 m) during the Triassic (Fig. 7b).

A further, discrete Triassic throw increase is observed along the eastern segment of the Fjerritslev South Fault, located between HF2 in the west and NS3 to the east (Fig. 3, 8). To the west of HF2, throw along the acoustic basement, BJU and top Jurassic horizons remains relatively constant at c. 500 ms TWT (c. 800 m), indicating a lack of pre-Early Cretaceous activity at this time (Fig. 8). However, between the HF2 and NS3 branchlines, throw increases to c. 700 ms TWT (1300 m), a 200 ms TWT increase above the BJU in this area, seemingly representing c. 200 ms TWT of Triassic throw in this area. Throw along the abutting HF2 and NS3 is similar, c. 200 ms TWT (500 m) (Fig. 3, 5). Fault displacement backstripping indicates that, west of the branchline with HF2, the Fjerritslev South Fault initiated during the Early Cretaceous, and that only the eastern segment was active during the Triassic (Fig. 8). An Early Cretaceous age for initiation of faulting along the western part of the Fjerritslev South Fault is further supported by our prior observation that Carboniferous-Permian, Triassic and Jurassic strata do not thicken across it (Fig. 2, 4).

Segments of the Fjerritslev North and South faults, located between branchlines with N-S striking faults, appear to have been active during the Triassic, at the same time as the N-S striking faults (Fig. 7b, 8). The main phase of activity along other parts of the E-W-striking faults occurred later, during the Early Cretaceous (Fig. 7b, 8b). One possible explanation for this relatively early, discrete Triassic activity could be that local segments of pre-



existing E-W-striking faults are reactivated between the N-S striking faults in response to an oblique stress field (cf. 'trailing segment reactivation' of Nixon et al., 2014). Although this model may explain the observed increase in throw along discrete E-W fault segments (Fig. 7a), with both N-S and E-W faults active simultaneously and accruing throw as a geometrically and kinematically linked system, it requires a pre-existing E-W fault that was subsequently reactivated as the trailing segment. This requirement is inconsistent with our seismic-stratigraphic evidence, which indicates that such a fault was not present prior to the Triassic (Fig. 2, 4).

We suggest that a more feasible hypothesis is that the distinctive, discrete throw increases relate to passive, post-formation lateral offset of the N-S striking faults due to sinistral strike-slip motion along E-W-striking faults. Such a model envisages juxtaposition of the hanging wall and footwall of a N-S striking fault across the E-W fault, with a geometric consequence being an increase in throw along the E-W fault in the absence of any dip-slip extension. This model does not require Triassic activity along E-W-striking faults, in agreement with the

Early Cretaceous timing of initiation for the Fjerritslev North and South faults (Fig. 7b, 8b). The local increases in throw would correspond to the difference in elevation between the hanging wall and footwall of the N-S-striking faults, which is essentially the throw accumulated on the N-S striking faults during the Triassic (Fig. 7a, 10). This model is supported by three observations. First, NS1 and NS2, and their respective hanging wall faults (HF1 and HF2), are geometrically similar, with HF1 and HF2 both showing folding and only minor offset of the

Acoustic Basement surface (Fig. 5, 9). Second, NS1 and NS2 are laterally offset by c. 10 km, as are HF1 and HF2, which are located in their respective hanging walls of the larger structures (Fig. 5, 9). Finally, NS1 and NS2 display similar values of throw at their branchlines with the Fjerritslev North Fault (c. 1000 ms TWT) to the discrete increase in throw that occurs between the intersections (Fig. 7a). This is consistent with the increase in throw being equivalent to the throw accrued along N-S faults during the Triassic.

Further evidence for sinistral strike-slip activity is observed at the northern margin of the basin, along the E-W-striking Farsund North Fault, implying offset of the NS1 and West Varnes Graben (WVG) fault (Fig. 1c, 10). A shallow footwall and deeper hanging wall, related to Early Cretaceous extensional activity, are identified along the Farsund North Fault, despite it being associated with a complex zone of deformation (Fig. 11). A key observation we make is that Carboniferous-Permian and Triassic strata are preserved in the footwall of the

Farsund North Fault, yet are eroded by the BJU in its hanging wall (Fig. 11). The footwall of the E-W Farsund North Fault also corresponds to the hanging wall of the N-S striking WVG fault, whereas the hanging wall of the Farsund North Fault corresponds to the footwall of the NS1 fault (Fig. 10b). We propose that, in the same way as we interpret to the south, the N-S striking WVG and NS1 faults initially represented a single, through-going structure, with erosion along its footwall occurring due to the BJU. This fault was then offset by c. 10 km

of sinistral strike-slip motion. This strike-slip activity juxtaposed the original hanging wall and footwall of the N-S fault across the Farsund North Fault, resulting in the complex and perhaps somewhat unusual structural and stratal geometries we observe (Fig. 11).

Having presented geometric observations showing strike-slip motions along E-W striking faults within the Farsund Basin, we now determine the timing of this activity. Based on the geometric and kinematic evidence

outlined above (Fig. 5, 7, 9, 11), we propose that the WVG, NS1 and NS2 faults initially formed a singular N-S striking fault during the Triassic, with HF1 and HF2, and HF2 and NS3 forming further, through-going, N-S





striking faults at this time. These faults were then offset along a series of E-W-striking, sinistral strike-slip faults, which appear to, in places, follow the location of, and may represent pre-cursors to, the present-day E-W-striking faults (Fig. 10b). This strike-slip activity must have occurred after the extensional activity along the N-S-striking faults during the Triassic (Fig. 6). Relatively smooth throw profiles and low displacement gradients along the Fjerritslev North and South faults on supra-BJU horizons (Top Jurassic and ILC; Fig. 7, 8), and the relatively isopachous Jurassic interval across the Farsund North Fault (Fig. 11), indicates that any strike-slip activity must have occurred prior to the deposition and preservation of Jurassic strata (Fig. 7, 8), most likely during the Early-Mid Jurassic, a period of either non-deposition or erosion by the BJU. More precise constraints on this age and evidence for any deformation associated with the activity are not possible given erosion along the BJU.

### 6.2 Late Jurassic-Early Cretaceous fault activity

Following Triassic extensional activity along N-S faults and strike-slip activity during the Early to Middle Jurassic, E-W-striking faults were active during the Late Jurassic-Early Cretaceous (Fig. 3, 6). Here we detail the geometric and kinematic behaviour of the Fjerritslev North and South faults in response to this tectonic event.

### 6.2.1 Fjerritslev North Fault

Fault displacement backstripping indicates that the Fjerritslev North Fault started to accommodate considerable amounts of extension during the Early Cretaceous (Fig. 7). At the ILC structural level, the central and western segments of the fault are expressed as a series of left-stepping en-echelon fault segments, with basinward-facing monoclines in their hanging wall (Fig. 12). Along the western segment of the fault, each individual en-echelon segment is 1-2 km long and strikes at c. 099°, a clockwise rotation of c. 11° from that of the overall E-W strike (088°) characterising the fault at deeper levels (Fig. 12). Along the central segment of the fault the en-echelon segments are each 3-5 km long (Fig. 12). The WNW-ESE-striking eastern segment lacks clear segmentation (Fig. 12).

En-echelon faults, geometrically similar to those observed along the Fjerritslev North Fault (Fig. 12), may form through the reactivation of a fault under a stress regime oblique to its orientation (Naylor et al., 1986; Richard, 1991; Grant and Kattenhorn, 2004; Swanson, 2006; Giba et al., 2012; Withjack et al., 2017) or pure dip-slip activity within mechanically anisotropic sequences (Schöpfer et al., 2007; Jackson and Rotevatn, 2013). However, based on the apparent lack of major lithological changes along-strike and the fact that the degree of en-echelon segmentation does change, we suggest that the fault geometry and the degree of obliquity experienced controls the degree of en-echelon segmentation rather than lithology. Grant and Kattenhorn (2004) show that oblique slip along a buried normal fault initially leads to the formation of a fault-parallel monoclinal fold at the surface. Further slip leads to fold breaching and preservation of a hanging wall monocline, with fault propagation associated with upward bifurcation of a single slip plane to form a series of en-echelon segments (see also Giba et al., 2012; and Withjack et al., 2017).



The strike of the Fjerritslev North Fault changes east of the branchline with NS2, from broadly E-W along the central segment (086°) to more WNW in the east, with this latter segment associated with a major depocentre (Fig. 12). Along the inside of the bend defined by this change in strike, WNW-striking faults, broadly parallel to the main structure, are observed within the footwall of the Fjerritslev North Fault (Fig. 3, 12). These faults are

geometrically similar to shortcut faults developed between differently oriented fault segments in the analog models of Paul and Mitra (2015). The outer bend (i.e. hanging wall) of the Fjerritslev North Fault is characterised by an array of small (c. 12 ms TWT, 15 m, throw) faults that strike NE, perpendicular to the main fault trace (Fig. 13), and which dip NW (Fig. 9). The faults are arranged into two main groups, situated at each

of the apexes of the fault bend (Fig. 13), with the larger faults situated in the east adjacent to a major hanging wall depocentre (Fig. 12, 13). These structures are, at last superficially, geometrically similar to 'hanging wall release faults' (*sensu* Destro, 1995; Stewart, 2001), which form as the hanging wall stretches along-strike so as to accommodate along-strike variations in fault displacement. However, we note that maximum throw on these faults occurs outboard of the main fault (Fig. 9, 13), and not at the branchline as would be expected for hanging

wall release faults. Instead, we propose that these hanging wall faults form in response to outboard stretching, accommodating extension around the convex bend in the fault plane, between the E-W-striking, more oblique, central and western fault segments, and the more optimally-oriented eastern segment (Fig. 12, 13).

### 6.2.2 Fjerritslev South Fault

Like the Fjerritslev North Fault, the Fjerritslev South Fault was active during the Early Cretaceous (Fig. 6c, 8).

However, a key observation is that, at the ILC structural level, the two faults differ markedly in their structural style; the Fjerritslev South Fault is more linear than the strongly segmented Fjerritslev North Fault (Fig. 12). The Fjerritslev South Fault is composed of several 5-10 km long strands, separated by footwall breached relay ramps and branchlines with pre-existing Triassic faults (Fig. 12). The presence of these breached relay ramps is not immediately obvious, although they can be inferred through the presence of composite monoclines (Fig. 12,

14). These composite monoclines comprise one fold hinge situated above the main fault trace, and a further fold hinge situated above the abandoned fault trace (Fig. 14). An additional composite monocline is present immediately west of the Fjerritslev South Fault's branchline with NS2, potentially indicating some segmentation and a relay ramp in this location.

The Fjerritslev South Fault initiated during the Early Cretaceous, accruing up to c. 500 ms (600 m) of throw

along its entire length (Fig. 8). West of HF2, no precursor fault is present, with the fault showing only Early Cretaceous activity (Fig. 4, 8). The apparent Triassic throw east of HF2 is attributed to an Early-Middle Jurassic stage of strike-slip activity with no Triassic dip-slip extension actually having occurred (Fig. 8b). Extensional activity along the Fjerritslev South Fault initiated during the Early Cretaceous, with the fault represented by a series of segments partitioned by pre-existing, N-S striking faults (NS2 and TF1, Fig. 12). An increase and

subsequent decrease in the relief of the acoustic basement surface across the footwall of the fault, to the east of NS2, may represent the eastern segment of the fault (Fig. 3). Further segments are observed west of TF1, separated by footwall breaching relay ramps and associated composite monoclines (Fig. 14).

In summary, based on: i) the left-stepping en-echelon segmentation observed along the western segment (Fig. 12); ii) the development of outer-bend faults in response to fault plane convexities associated with changes in



strike (Fig. 13); and iii) the development of a major depocentre within the hanging wall of the WNW-striking eastern segment, we propose that the Fjerritslev North Fault was obliquely reactivated under a dextral transtensional stress regime during the Early Cretaceous (Fig. 15c). The western part of the Fjerritslev South Fault initiated as a new structure at this time under the same dextral transtensional stress regime, propagating westwards away from the prior strike-slip fault initially constrained between HF2 and NS3 (Fig. 10c, 15c).

**7 Discussion**

**7.1 Geometric and kinematic development of multiphase rift-related fault networks during non-coaxial extension**

We have determined the geometric and kinematic evolution of upper-crustal faults within the Farsund Basin. Triassic N-S striking faults were offset by a series of E-W-striking, sinistral strike-slip faults during the Early-

Mid Jurassic (Fig. 10a, b, 15a). Transtensional strike-slip systems in nature and in experimental models often display anastomosing, duplex geometries, where strain is distributed along a series of discrete structures that link in both strike and dip directions (e.g. Naylor et al., 1986; Richard et al., 1995; Schreurs, 2003; Vauchez and Tommasi, 2003; Mann, 2007; Wu et al., 2009; Scholz et al., 2010; Dooley and Schreurs, 2012; Chamberlain et al., 2014; Corti and Dooley, 2015; Cheng et al., 2017). However, due to BJU erosion, we can only confirm prior

strike-slip activity between the laterally offset N-S striking faults, i.e. NS1 and NS2 (Fig. 7, 10), HF2 and NS3 (Fig. 8, 10), and WVG and NS1 (Fig. 10c, 11). Where we know strike-slip faulting occurred, i.e. between the offset N-S faults, the strike-slip faults correspond to the same location as Early Cretaceous normal faults, i.e. the Fjerritslev North and Farsund North faults (Fig. 15b, c), observed at the present-day, potentially indicating that these later extensional faults reactivated the prior strike-slip ones. However, the Fjerritslev South Fault west of

HF2 cannot have reactivated an older strike-slip fault. The strike-slip fault offsetting HF2 and NS3 does not share the same location as the Fjerritslev South Fault west of HF2, as the lack of offset of NS2 would imply an unrealistically steep displacement gradient (Peacock and Sanderson, 1991) (Fig. 15b). Instead, we propose that, in this location, the strike-slip fault strikes NW-SE, following the location of HF2 (Fig. 15b), and joining with the strike-slip fault between NS1 and NS2 (later reactivated as the Fjerritslev North Fault).

Fault displacement backstripping (Fig. 7, 8) and seismic stratigraphic observations (Fig. 2, 4) indicate an Early Cretaceous onset of extension for the E-W striking faults. This is consistent with our interpretation of Early-Middle Jurassic strike-slip activity along E-W-striking faults, with these faults experiencing no dip-slip motions prior to the Early Cretaceous. Apparent Triassic throw along the central segment of the Fjerritslev North Fault and the eastern segment of the Fjerritslev South Fault occurred through the passive juxtaposition of different

structural levels across the fault, as opposed to active dip-slip faulting. This study shows that, particularly in areas of non-colinear faulting and oblique stress fields, care needs to be taken when interpreting fault kinematics from T-x profiles alone. In such instances, further lines of evidence, such as the presence of syn-kinematic strata, are required to confirm extensional fault activity.

En-echelon fault segmentation, such as observed along the Fjerritslev North Fault at the ILC structural level

(Fig. 12), may form due to the oblique reactivation of a pre-existing fault (Grant and Kattenhorn, 2004; Giba et al., 2012; Lăpădat et al., 2016; Withjack et al., 2017). In the case of the Fjerritslev North Fault this is likely to be





the pre-existing strike-slip fault (Fig. 10c, 15). Conversely, the Fjerritslev South Fault, which was also active during the Early Cretaceous, is more linear and not obviously segmented. In addition, its western segment does not have the same strike as the pre-existing strike-slip fault (Fig. 15b). The Fjerritslev South Fault instead

appears to propagate westwards away from the prior strike-slip fault in the east as a newly formed structure, rather than simply reactivating a pre-existing one, hence the difference in structural style between it and the Fjerritslev North Fault (Fig. 8, 15). Geometrically, the Early Cretaceous basin-bounding Fjerritslev South Fault, propagating away from the pre-existing strike-slip fault, resembles the lateral propagation of faults bounding pull-apart basins (Mann et al., 1983; Dooley and Schreurs, 2012; Corti and Dooley, 2015). The Farsund North

Fault, which also shows only Early Cretaceous extensional activity (Fig. 2, 11), appears to show a complementary eastwards propagation along the northern margin of the basin (Fig. 15c). The geometry and kinematic history of the Fjerritslev North Fault during the Early Cretaceous, in particular the development of en-echelon segmentation and formation a pull-apart basin in the more optimally oriented eastern fault segment, also broadly resemble faults formed within pull-apart basins during dextral transtension (Naylor et al., 1986;

Richard, 1991; Richard and Krantz, 1991; Richard et al., 1995; Grant and Kattenhorn, 2004). More regionally, the crustal-scale fault system defining the Farsund Basin resembles a negative flower structure (Cheng et al., 2017) (Fig. 2). Based on the evidence outlined above, we propose that the Farsund Basin formed as a pull-apart basin in response to Early Cretaceous dextral transtension. In the east of the basin, rapid subsidence may locally have been accentuated by salt mobilisation (Christensen and Korstgård, 1994).

**7.2 Relation of the STZ to the regional tectonic setting**

We have determined the kinematic history of upper crustal faults during multiple tectonic events, which we propose link to, and reflect activity along, the STZ at sub-crustal depths. Here, we examine the driving forces behind these tectonic events. Due to the fixed E-W orientation of the upper-crustal expression of the STZ relative to later tectonic events, we are able to use the observed reactivation style to infer the prevailing regional

stress field at that time.

During the Carboniferous-Permian, dextral transtension and transpression occurred on a series of NW-trending faults along the STZ and TTZ (Fig. 16a) (e.g. Liboriussen et al., 1987; Michelsen and Nielsen, 1993; Mogensen, 1994; Erlström et al., 1997), including along the Fjerritslev Fault system to the east of the basin (Hamar et al., 1983; Skjerven et al., 1983; Mogensen, 1994). We document no such activity in the Farsund Basin (Fig. 2, 4).

Sinistral motion is also proposed along the Skagerrak Graben at this time (Fanavoll and Lippard, 1994; Lie and Husebye, 1994; Mogensen, 1994), which, when combined with dextral motion along the STZ, results in net E-W extension, accommodating N-S compression from the Variscan Orogeny to the south (Fig. 16a). As such, we may not expect any extensional activity along the E-W faults within the Farsund Basin. N-S-striking faults in and around the Farsund Basin do not appear to be related to the STZ; instead, these faults appear to form the

northern continuation of the Horn Graben, which formed in response to Permian-Triassic E-W-directed extension (Vejbæk, 1990; Nielsen, 2003).

Throughout the Mesozoic, both sinistral (Pegrum, 1984; Liboriussen et al., 1987; Norling and Bergström, 1987; Sivhed, 1991; Jones et al., 1999) and dextral (Michelsen and Nielsen, 1993; Mogensen, 1995; Erlström et al., 1997; Hansen et al., 2000; Mogensen and Korstgård, 2003; Bergerat et al., 2007; Graversen, 2009) strike-slip



and oblique motions have been documented at various locations along the STZ. Sinistral strike-slip activity along an E-W trending structure, such as observed during the Early-Middle Jurassic (Fig. 16c), could be driven by regional E-W to NW-SE oriented extensional stress fields. Potential regional driving mechanisms during the Early-Middle Jurassic include: i) localised extension north of the STZ within the Skagerrak Graben (Ro et al., 1990; Vejbæk, 1990); ii) regional stresses relating to the inflation of the Mid North Sea thermal dome (Rattey

and Hayward, 1993; Underhill and Partington, 1993); or iii) far-field stresses relating to the opening of the Atlantic and Tethyan oceans to the south (Ziegler, 1990; Ziegler and Stampfli, 2001; Vissers et al., 2013). The Skagerrak Graben was inactive throughout the Jurassic and was furthermore decoupled from the STZ at this time (Ro et al., 1990); accordingly, localised extension along the Skagerrak Graben could not have driven the observed strike-slip activity. The influence of the Mid North Sea thermal dome was felt in the Farsund Basin

during the Early-Middle Jurassic and associated stresses would be localised south of, and potentially buffered by, the STZ (Underhill and Partington, 1993) (Fig. 16c). Although it is difficult to envisage how stresses arising from broad, regional inflation result in strike-slip activity, this still may represent a possible explanation, particularly if the uplift is buffered at the STZ (Fig. 16c). A further possibility is that the strike-slip activity is related to far-field stresses transmitted along the Tornquist Zone associated with ocean formation to the south

(e.g. Ziegler and Stampfli, 2001; Coward et al., 2003; Vissers et al., 2013).

Around the Early-Middle Jurassic, the incipient Piemont-Ligurian Ocean was situated south of the STZ, between the northwards subducting Tethys to the east and the Central Atlantic to the west (Ziegler, 1990; Ziegler and Stampfli, 2001; Coward et al., 2003; Vissers et al., 2013). Seafloor spreading within the Piemont-Ligurian Ocean initiated at around 170 Ma, during the Mid-Jurassic, with the ocean basin and prior rifting

buttressed to the east at the TTZ (Vissers et al., 2013). Continental extension occurred during the Early-Mid Jurassic, prior to seafloor-spreading, and was associated with sinistral strike-slip motion along the TTZ (Vissers et al., 2013) (Fig. 16). Compressional forces arising from the subduction of the Tethys to the west may have also led to sinistral strike-slip motions being transmitted along the NW trending Tornquist Zone, potentially reaching the Farsund Basin (see Fig. 7 in Ziegler, 1990) (Fig. 16c). The Tornquist Zone may have acted as a proto-

transform structure, representing an ultimately failed link between the North Atlantic rift system to the northwest and the Tethys to the southeast, similar to the transform fault between Iberia and Central Europe (see Fig. 7 in Vissers et al., 2013) (Fig. 16c). One implication of this model is the occurrence of sinistral motion along the whole of the Tornquist Zone at this time, a requirement not necessitated by the thermal dome hypothesis (Fig. 16c). Based on our data, we are unable to comment on whether this activity occurred along the

structure to the east, although some sinistral activity is observed to the west (Skjerven et al., 1983; Pegrum, 1984; Jones et al., 1999).

Dextral transtensional activity occurred within the Farsund Basin during the Early Cretaceous (Fig. 12, 13, 15), as observed elsewhere along the STZ at this time (Michelsen and Nielsen, 1993; Mogensen and Jensen, 1994; Bergerat et al., 2007). Based on the easterly trend of the STZ in this location, the driving regional stress field

could be extension and orientated approximately E-W to NE-SW (Fig. 15). Regional extension occurred across the North Sea during the Late Jurassic-Early Cretaceous and was oriented E-W to NE-SW in the Central and Southern North Sea (Rattey and Hayward, 1993; Underhill and Partington, 1993; Coward et al., 2003), and E-W to NW-SE in the Northern North Sea (Bartholomew et al., 1993; Brun and Tron, 1993; Bell et al., 2014).



Dextral transtension and the formation of a pull-apart basin would indicate that NE-SW-to-E-W extension, associated with activity within the Central Graben, was prevalent at this time within the Farsund Basin (Fig. 16d).

**7.3 Nature and reactivation of lithospheric lineaments**

We have shown that the upper crustal part of the STZ was repeatedly reactivated in a range of different tectonic styles, which we have linked to regional tectonic events, prevailing stress fields, and potentially broader geodynamic context (Fig. 16). Here, we discuss how such a contrast in lithospheric thickness and properties at depth (e.g. Kind et al., 1997; Cotte and Pedersen, 2002; Hossein Shomali et al., 2006), is able to influence rift development within the upper crust.

Changes in lithospheric thickness associated with prior phases of rifting or different continental blocks have previously been shown to influence the development of rift systems (Corti, 2009; Autin et al., 2013; Brune et al., 2017). In addition, numerous studies show that strike-slip and oblique fault systems can dissect the whole lithosphere and are often associated with pervasive fabrics within mantle lithosphere (Wylegalla et al., 1999; Tommasi and Vauchez, 2001; Vauchez and Tommasi, 2003). Examples from the Great Glen Fault, UK (Klemperer and Hobbs, 1991; McBride, 1995), the Transbrasiliano fault zone, onshore Brazil (Daly et al., 2014), and the San Andreas fault system, USA (Chamberlain et al., 2014), represent crustal-terrane separating fault systems that extend down to at least the base of the crust and, in many cases, into sub-crustal lithosphere (Vauchez and Tommasi, 2003). These lithosphere-scale structures are often oriented oblique to regional tectonic events and are thus subjected to oblique stresses; as such, they are often reactivated in a transpressional or transtensional manner, resulting in complex rifts at upper crustal levels (e.g. Underhill and Brodie, 1993; Holdsworth et al., 2001; Vauchez and Tommasi, 2003; Bergerat et al., 2007; Le Breton et al., 2013; Corti and Dooley, 2015; Calignano et al., 2017; Cheng et al., 2017).

The E-trending Farsund Basin and STZ represent a further example of a lithosphere-scale structure (Fig. 2). Lassen and Thybo (2012) propose the STZ formed as an Ediacaran rift system during the breakup of Rodinia. Montalbano et al. (2016) note a sinistral phase of motion along the present-day trend of the STZ around this time, indicating that the structure must have existed in some form at this time. Although the STZ appears to have been present in some form since the Late Proterozoic (Lassen and Thybo, 2012; Montalbano et al., 2016), its upper crustal component appeared to have been established during the Carboniferous-Permian as a lithosphere-scale strike-slip system, exploiting the change in lithospheric thickness and properties at depth (Erlström et al., 1997; Lassen and Thybo, 2012). The localised reactivation of this structure appears to buffer the cratonic lithosphere of Baltica against the regional tectonic events that affect the amalgamated terranes of Central Europe (cf. Berthelsen, 1998; Mogensen and Korstgård, 2003). Berthelsen (1998) propose that the Fjerritslev Fault System forms a NE-dipping detachment that shields the cratonic lithosphere of Baltica. Although our data are unable to confirm details of this hypothesis, we do note a connection between the sub-crustal and upper crustal components (Fig. 2), which may fulfil the role of buffering Baltica. As the orientation of this lithosphere-scale structure is fixed relative to the later tectonic events (Cotte and Pedersen, 2002; Babuška and Plomerová, 2004), yet is weak enough to be repeatedly reactivated throughout these events, seemingly regardless of orientation, the style of reactivation of the upper crustal component can be inverted to



better understand the regional stress field that has affected the North Sea throughout these multiple tectonic
events.

**8 Conclusions**

In this study we use the geometric and kinematic evolution of a complex upper crustal fault population to better
understand the kinematic behaviour of a linked deeper structure, the lithosphere-scale Sorgenfrei-Tornquist
Zone. We find that:

1. The E-W faults that define the margins of the Farsund Basin form a linked system both laterally and at depth.
These represent a crustal-scale fault system and are interpreted to extend through the Moho to the change in

lithospheric thickness and properties representing the Sorgenfrei-Tornquist Zone within the sub-crustal
lithosphere. Together, this lithospheric step and the upper-crustal rift system form the lithosphere-scale STZ.

2. The southern margin of the Farsund Basin is characterised by N-S- and E-W-striking fault populations. No
activity is observed within the Farsund Basin during Carboniferous-Permian activity, in contrast to elsewhere
along the STZ. Extension across N-S-striking faults occurs in the Triassic, with E-W-striking faults beginning to

accommodate extension during the Late Jurassic-Early Cretaceous.

3. Sinistral strike-slip motion is observed along a number of E-W-striking faults during the Early-Mid Jurassic,
acting to offset (c. 10 km) pre-existing N-S faults. This results in the juxtaposition of the hanging wall and
footwall of these pre-existing faults, creating apparent throw along the fault, without any extension having
occurred. This represents a previously undocumented phase of activity across the North Sea, and may be linked

to far-field stresses arising from the opening of oceanic basins to the south, or the inflation of the Mid North Sea
thermal dome to the west.

4. Sinistral strike-slip activity is succeeded during the Early Cretaceous by dextral transtension, as evidenced
through the oblique reactivation and lateral propagation of E-W-striking faults. This phase of activity is linked
to the regional NE-SW oriented regional Late Jurassic-Early Cretaceous extension and resulted in the formation

of the present-day morphology of the Farsund Basin as a transtensional pull-apart basin.

5. The lithosphere-scale STZ represents a long-lived lithospheric weakness that is preferentially reactivated in
an oblique manner during later tectonic events, with the style of reactivation being dependent on the regional
stress field. The observed style of reactivation can offer insights into the prevailing stress field at various points
throughout the protracted history of the STZ, in some cases highlighting previously unknown tectonic events.

6. We find that structures within the sub-crustal lithosphere are often associated with complex upper crustal rift
systems and may exert a strong influence over their geometry and development. Regional stress fields at oblique
angles to these structures may result in transpressional or transtensional activity along structures within the
upper crust. The geometric and kinematic evolution of faults populations within these upper crustal structures
are not only able to offer insights into the kinematic behaviour of the structures within the sub-crustal

lithosphere, but are also inherently linked to the larger-scale regional stress field at the time.



**Author contributions**

The seismic interpretation and analyses throughout this study were undertaken by TP. Interpretations and evolutionary models were arrived at by TP, with additional input by CJ, RB and OD. The manuscript was written by TP, with additional input and scientific editing from CJ, RB and OD. All authors contributed to
extensive discussions and ideas throughout the study and the writing of the manuscript.

**Acknowledgements**

This contribution forms part of the MultiRift Project funded by the Research Council of Norway's PETROMAKS programme (Project number 215591) and Statoil to the University of Bergen and partners Imperial College, University of Manchester and University of Oslo. The authors would like to thank PGS for providing and allowing
us to show the seismic data used throughout this study. We would also like to thank Schlumberger for providing academic licences of the Petrel software to Imperial College. In addition, we thank members of the Basins Research Group, in particular Alex Coleman and Matthew Reeve, for valuable discussions throughout this study.

**Appendices**

*Appendix A – Throw-length fault analyses*

Throw measurements were taken every 150m along strike of key faults at various stratigraphic horizons to create a series of Throw-length (T-x) plots. These plots record vertical displacement along the fault plane for a specified stratigraphic horizon, and as such makes no inference as to how this throw accumulated, i.e. dip-slip/oblique-slip.

To accurately constrain the evolution of a fault all fault slip-related strain much be explicitly recorded; this
means that ductile deformation, such as fault-parallel folding, and brittle strains associated with fault displacement must be incorporated into the throw-length plot (Meyer et al., 2002; Long and Imber, 2010; Whipp et al., 2014; Duffy et al., 2015). Where fault-parallel folding occurs, hanging wall and footwall cut-offs were defined by projecting the regional dip of the horizon, as measured some distance away from the fault, to the fault plane (see supplementary Fig. 1). The discrepancy between projected (i.e. brittle plus ductile) and non-
projected (i.e. brittle) throw measurements represents deformation accommodated via ductile means (e.g. folding) (e.g. Long and Imber, 2010; Duffy et al., 2015). Footwall erosion occurs across some of the faults in the area, meaning the cut-offs of some stratigraphic horizons are absent (Fig. 1c). To constrain throw in these cases, and remove the effects of this erosion, we project the regional trend of the horizon, as measured from its subcrop below the erosional unconformity updip, towards a projection of the fault plane (see supplementary Fig. 1).

*Appendix B – Fault-displacement backstripping*

T-x plots were calculated across multiple stratigraphic horizons for each of the major faults. At each point along the fault, by systematically removing throw accrued during specific stratigraphic intervals, beginning with the youngest, we are able to determine the throw distribution along the fault at various points back in time. As such, we can also determine the lateral extent and any segmentation along the fault back in time, where backstripped
throw is equal to zero, we can assume that the fault was not present in that location at the specified time. Due to



independent constraints, obtained from isochron analysis, on the lateral extent of the fault at various points in its evolution, we use the 'vertical throw subtraction' backstripping method (Chapman and Meneilly, 1991; Childs et al., 1993; Tvedt et al., 2016) as opposed to the modified 'T-max method' (Rowan et al., 1998; Dutton and Trudgill, 2009) (see discussion by Jackson et al., 2017).






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



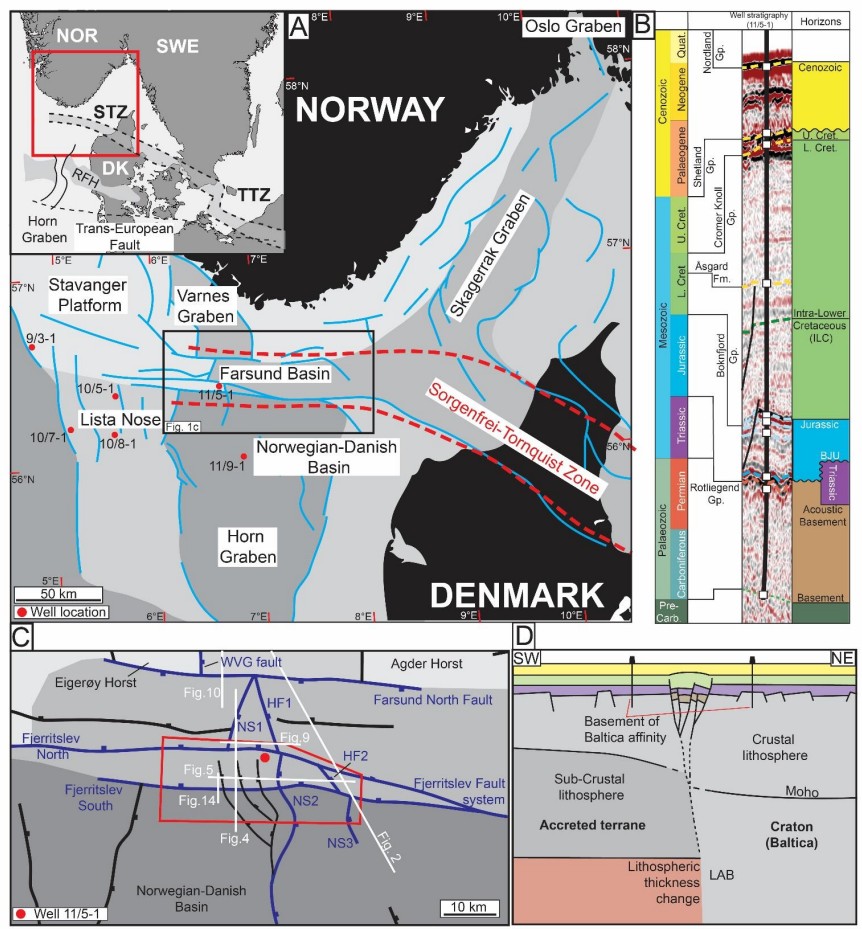

**Figure 1.** A) Regional map of the study area showing the relation to the major structural elements and fault networks. Well locations used to constrain ages of stratigraphic horizons are shown in red. Inset – Regional map of the area showing the location and geometry of the Tornquist Zone, along with major crustal-scale structural elements. B) Stratigraphic column showing the stratigraphy encountered in well 11/5-1 (located within the 3D volume), and the major tectonic events to have affected the region. C) Map showing the faults networks present across the Farsund Basin (those referred to in the text shown in blue), the location of the 3D seismic volume, and the main figures used throughout the study. D) Schematic cross-section showing the Sorgenfrei-Tornquist Zone as defined at both upper crustal and sub-crustal depths, and the potential relationship between the two.





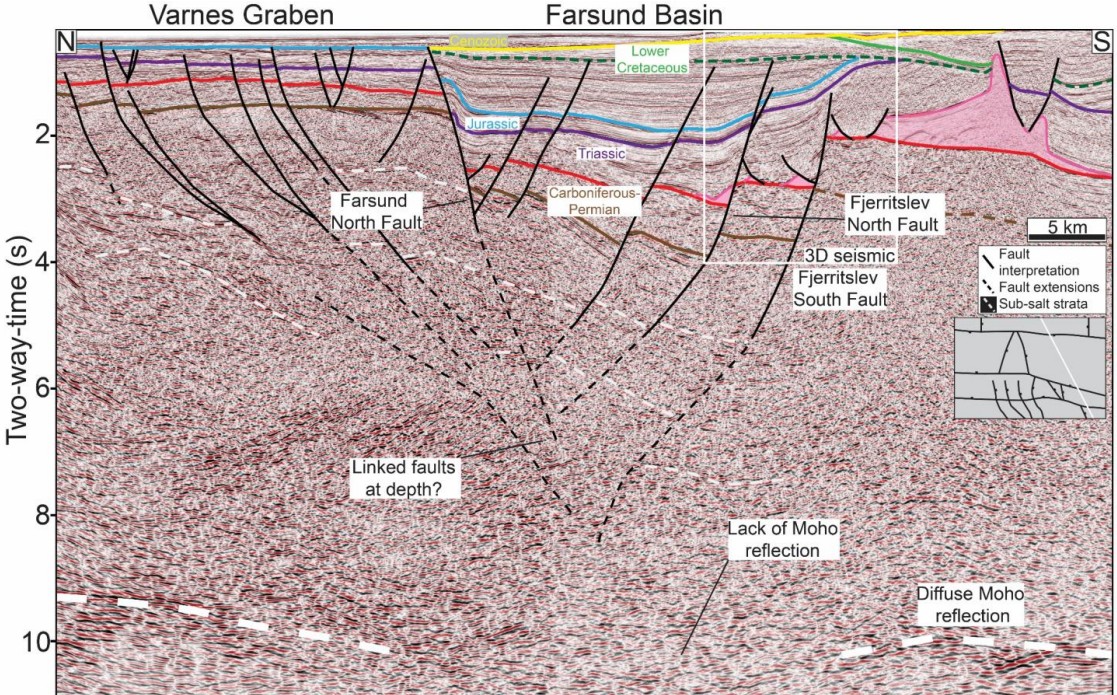

**Figure 2.** Interpreted N-S oriented seismic section across the Farsund Basin showing the linked and crustal-scale nature of the E-W oriented Basin bounding faults. The lack of Moho reflection directly beneath the basin may imply that the fault cross-cut the Moho in this area and extend through the crust. See Fig. 1c for location.



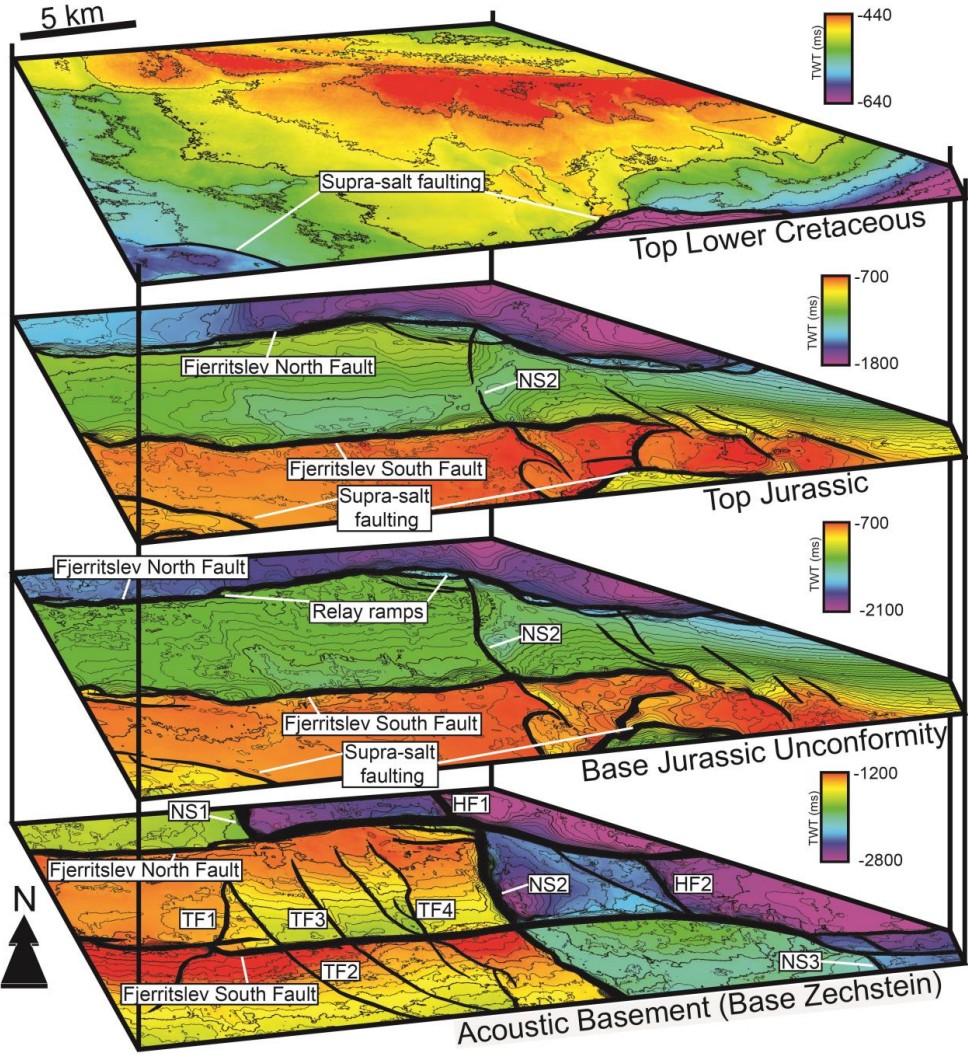

**Figure 3.** Two-way-time structure maps of the key stratigraphic horizons used within this study, as mapped within the 3D seismic volume. The acoustic basement (Base Upper Permian Zechstein salt), Base Jurassic Unconformity, Top Jurassic, and Top Lower Cretaceous surfaces are shown. Vertical separation not to scale. See Fig. 1c for location of the 3D seismic volume.



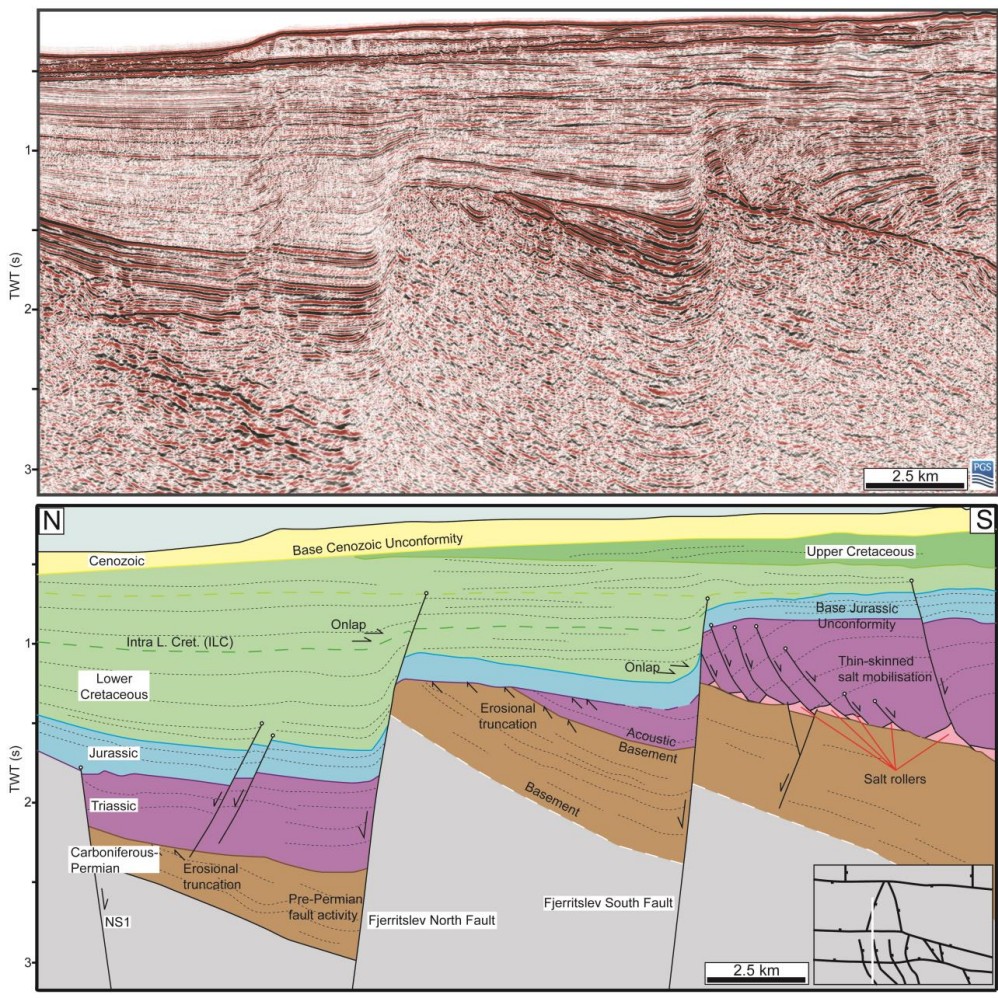

**Figure 4.** Uninterpreted and interpreted N-S oriented seismic section across the Farsund Basin. The Fjerritslev South Fault appears to show solely Early Cretaceous activity, with some apparent Triassic activity preceding Early Cretaceous activity along the Fjerritslev North Fault. No pre-Permian activity is apparent across either fault. See Fig. 1c for location.



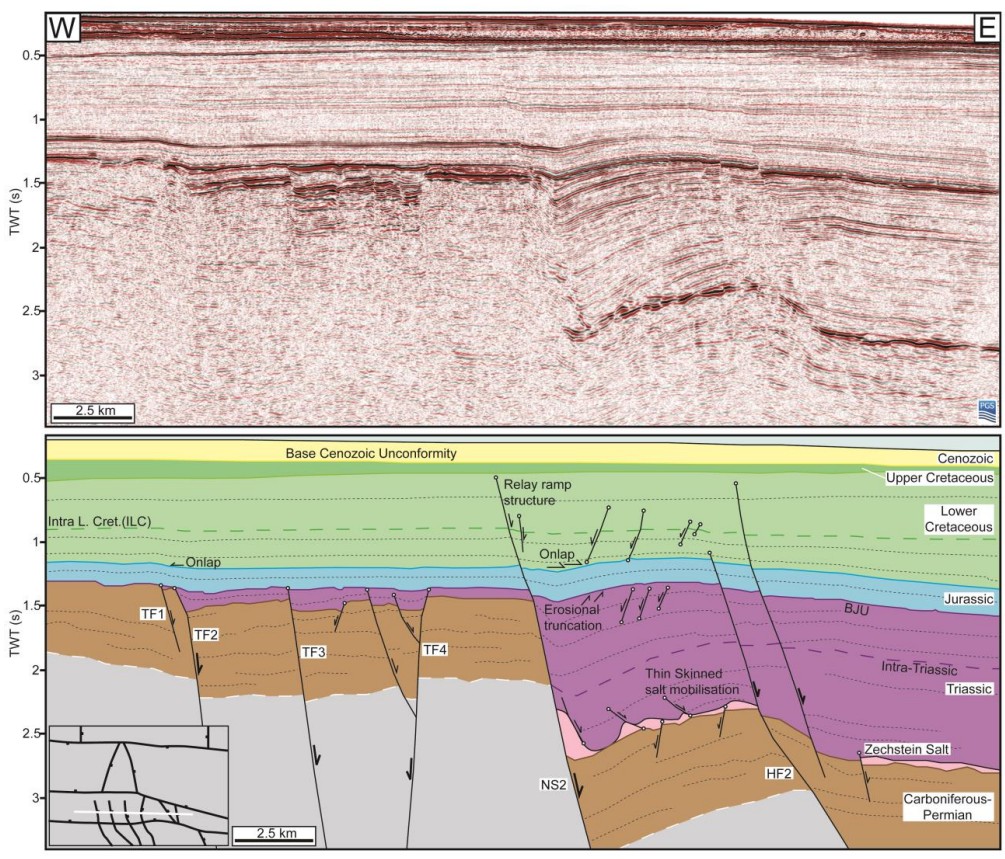


**Figure 5.** Uninterpreted and interpreted E-W oriented seismic section across the Farsund Basin. Triassic thickness changes are observed across N-S striking faults, with other stratigraphic intervals appearing largely isopachous. See Fig. 1c for location.



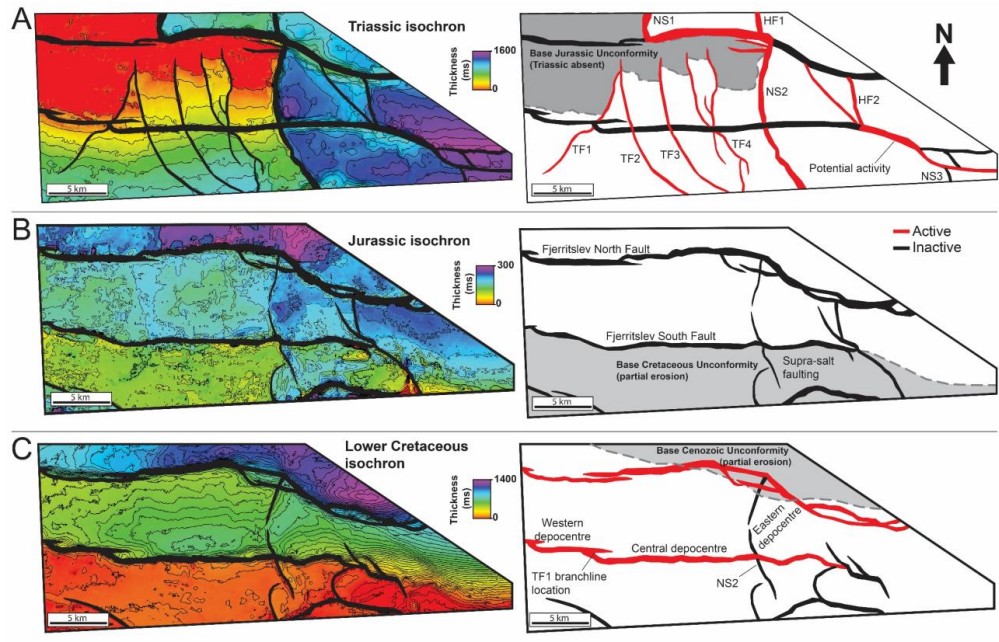

**Figure 6.** A) Isochron showing thickness of the Triassic interval and the associated faults active during this interval based on sediment thickness changes and depocentres. B) Isochron of Jurassic strata showing thickness and the faults that appear to have been active during this interval. C) Isochron of Lower Cretaceous strata showing the faults that appeared to be active during this interval.



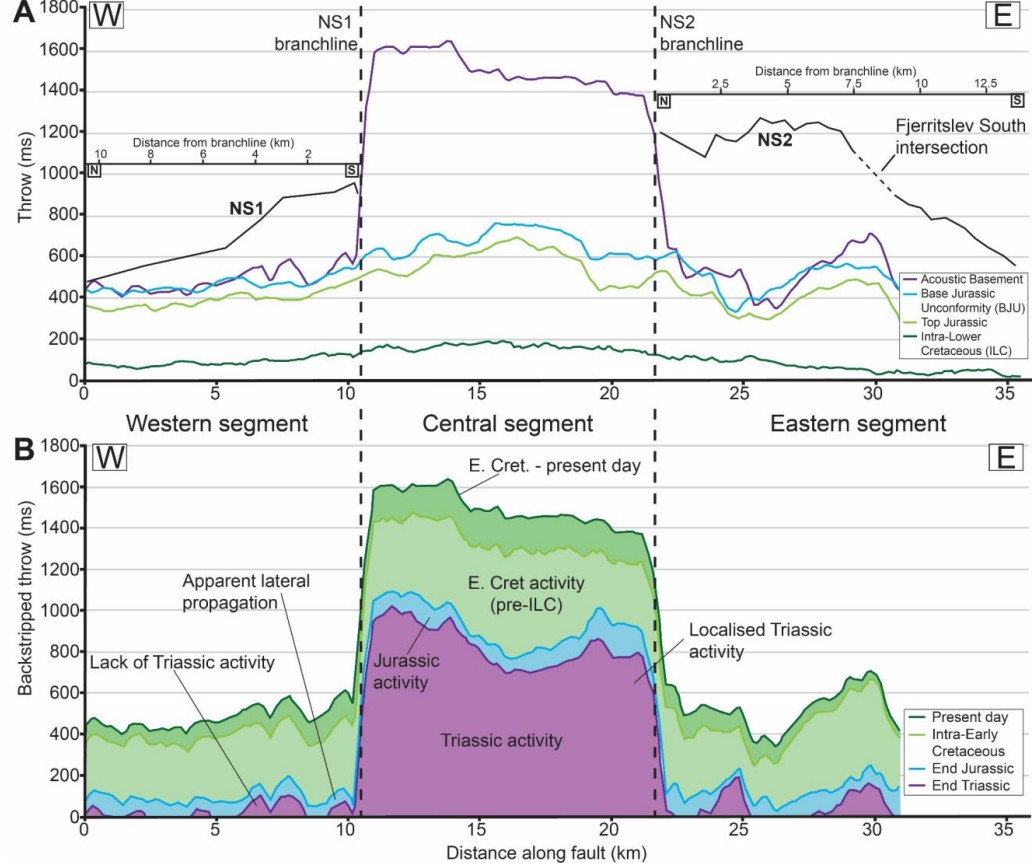

**Figure 7.** A) Throw-length profiles calculated for major stratigraphic horizons along the Fjerritslev North Fault showing the branchlines with other faults. Shown in black are the throw-length profiles for the NS1 and NS2 faults calculated across the acoustic basement horizon. Note the correspondence between the NS1 and NS2 fault branchlines and the marked increase in throw along the acoustic basement horizon. B) Backstripped fault profile for the Fjerritslev North Fault showing the kinematic evolution of the fault. The central segment accommodated throw during the Triassic, before the rest of the fault became active and throw accumulated along the length of the fault during the Early Cretaceous.




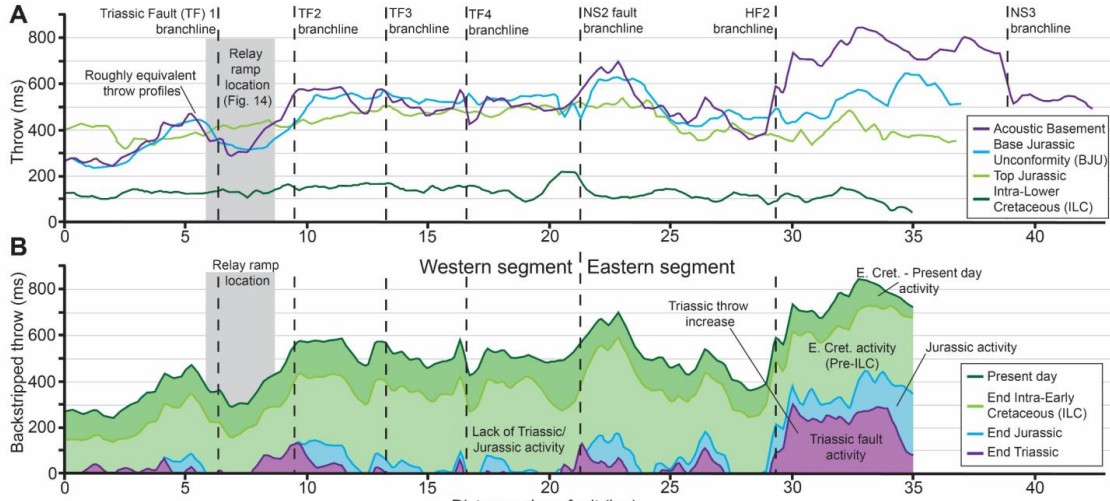

**Figure 8.** A) Throw-length profiles calculated for major stratigraphic horizons across the Fjerritslev South Fault showing the branchlines with cross-cutting faults. Note the similarity between the different stratigraphic horizons west of the HF2 branchline. B) Backstripped fault profile for the Fjerritslev South Fault showing Triassic activity east of the HF2 branchline but not elsewhere along the fault. Early Cretaceous fault activity is observed along the length of the fault.



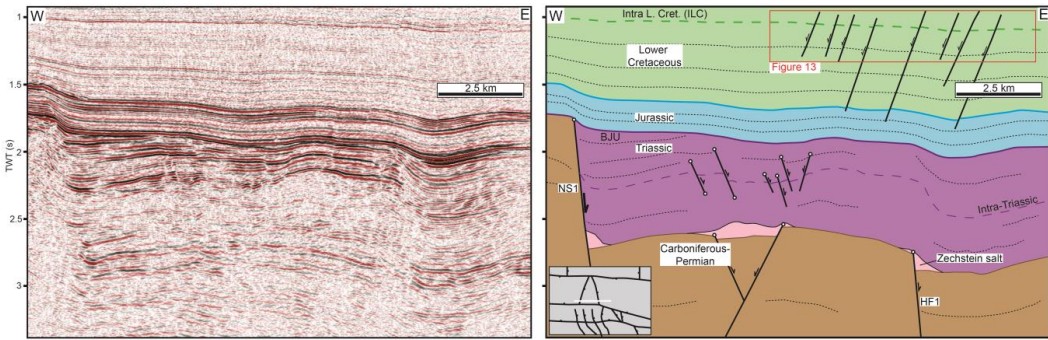

**Figure 9.** Interpreted and uninterpreted E-W oriented seismic section located across NS1 and within the hanging wall of the Fjerritslev North Fault. Triassic activity occurs across NS1, with little Early Cretaceous activity observed. Note the geometric similarities between NS1 and HF1 on this section, and NS2 and HF2 on Fig. 5. See Fig. 1c for location.




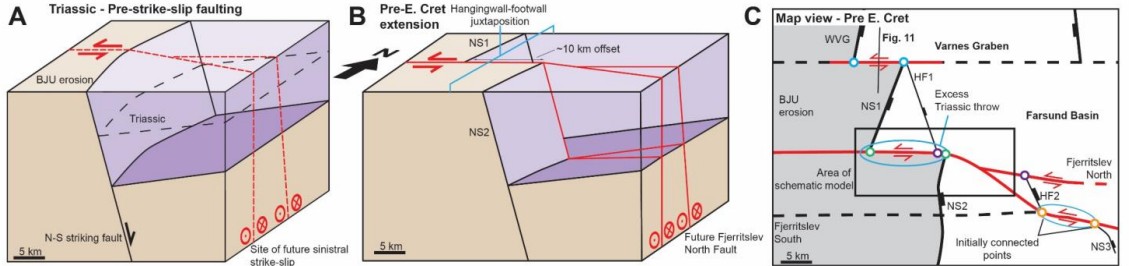

**Figure 10.** A) Schematic diagram showing Triassic activity along N-S striking faults. B) Schematic diagram showing the resultant fault geometries and associated hanging wall-footwall juxtaposition as a result of the sinistral offset of the N-S striking fault. C) Map showing the geometry of the strike-slip system within the Farsund Basin prior to the Early Cretaceous. Black box shows the location of the schematic model showing hanging wall-footwall juxtaposition, with blue ovals showing areas of excess Triassic throw (See Fig. 7, 8).





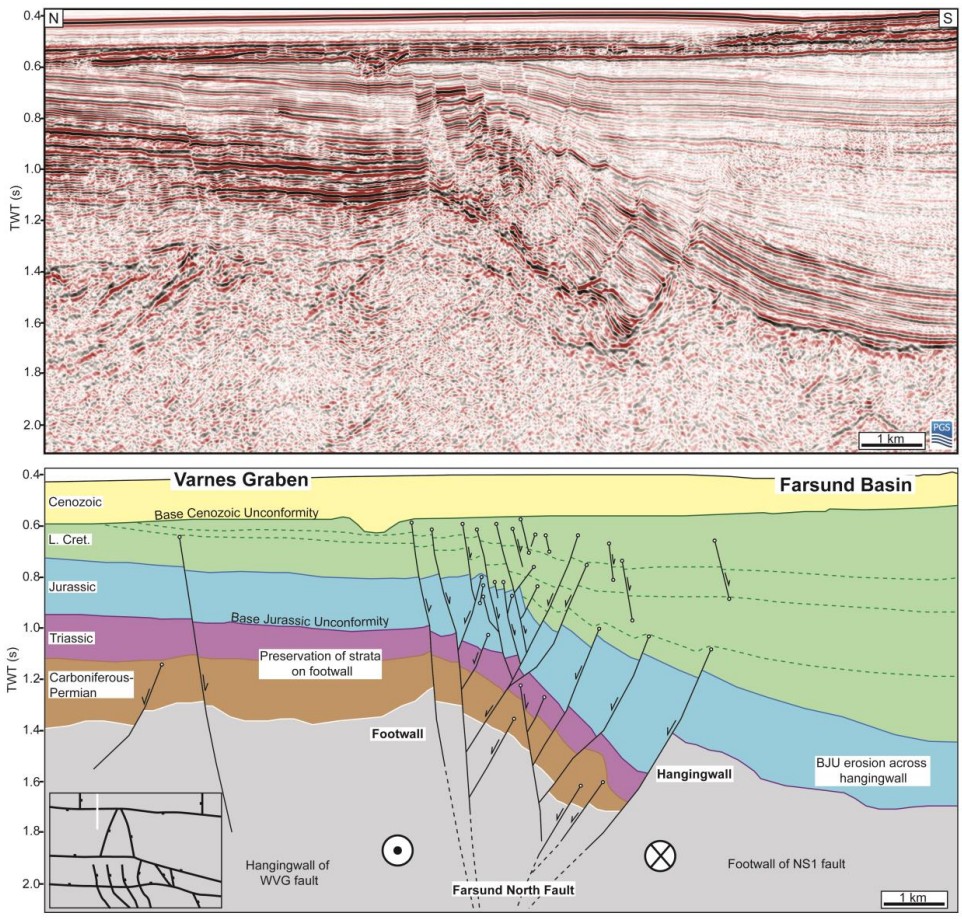

**Figure 11.** Interpreted and uninterpreted N-S oriented seismic section across the Farsund North Fault.
Preservation of Carboniferous-Permian and Triassic strata across the footwall of the Farsund North Fault
with concomitant erosion across the hanging wall represents the same hanging wall-footwall juxtaposition
demonstrated in Fig. 10. See Fig. 1c, 10 for location.



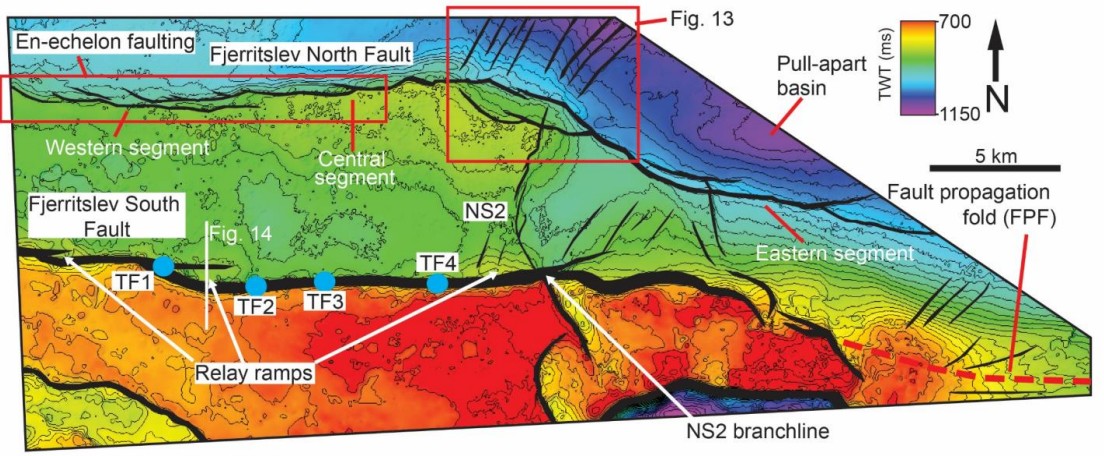

**Figure 12.** Two-way time structure map of the Intra-Lower Cretaceous horizon (see Fig. 4, 5 for surface in

section). The Fjerritslev North Fault shows clear en-echelon segmentation along its western segment and a

deepening of the surface in its hanging wall to the northeast. Intersections of underlying Triassic N-S striking

faults with the Fjerritslev South Fault at deeper stratigraphic levels are indicated by blue circles.

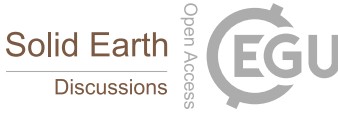



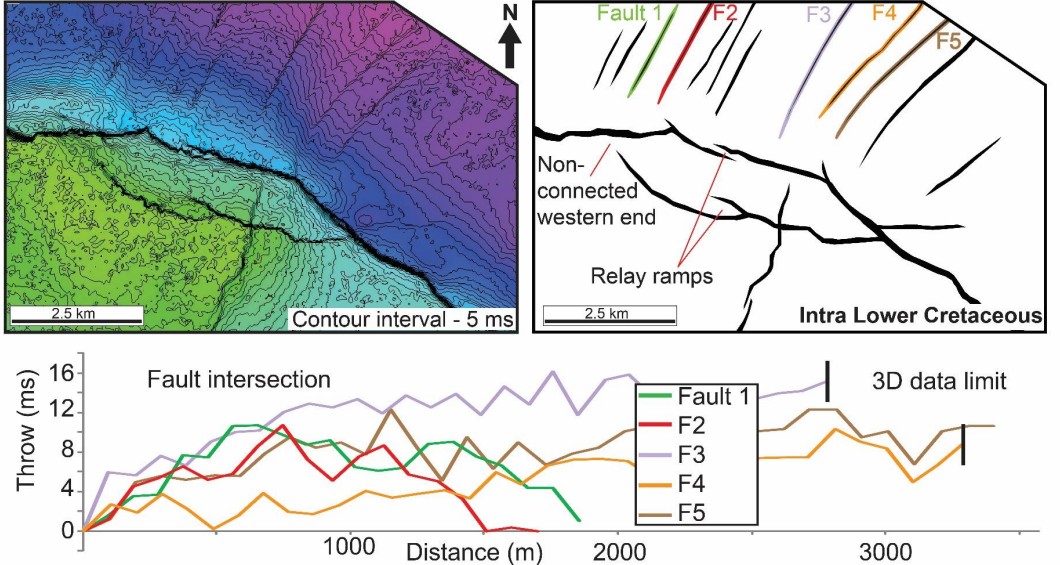

**Figure 13.** A) Interpreted and uninterpreted fault geometries of the bend in the Fjerritslev North Fault at the ILC stratigraphic horizon. A series of minor, NE-striking faults are present around the outside of the bend, perpendicular to the main fault trace. B) T-x profiles for the outer-bend faults as calculated for the ILC stratigraphic horizons. Throw maxima are observed outboard of the main Fjerritslev North Fault trace.



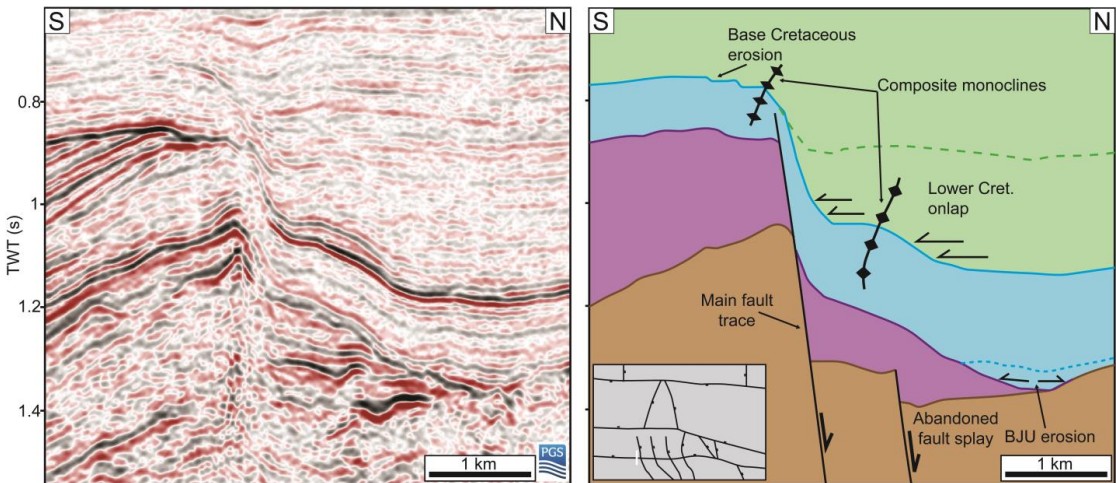

**Figure 14.** Interpreted and uninterpreted N-S oriented seismic section across the Fjerritslev South Fault. Section shows the presence of a composite monocline and associated underlying relay ramp.




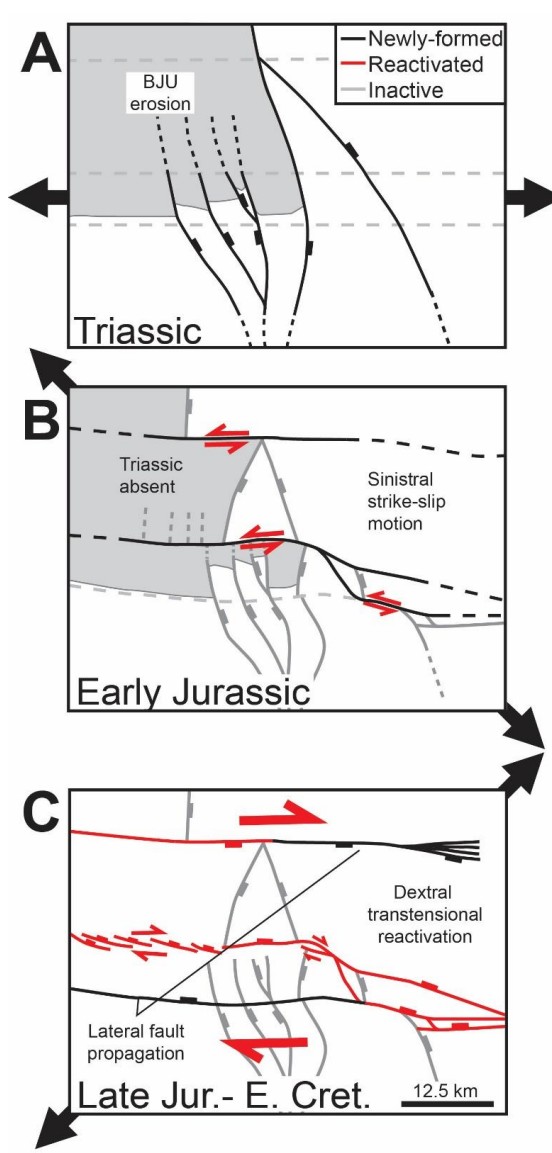

**Figure 15.** Schematic model showing the fault geometries and kinematics during A) Triassic E-W extension; B) Early-Mid Jurassic sinistral strike-slip activity; and C) Early Cretaceous dextral transtension




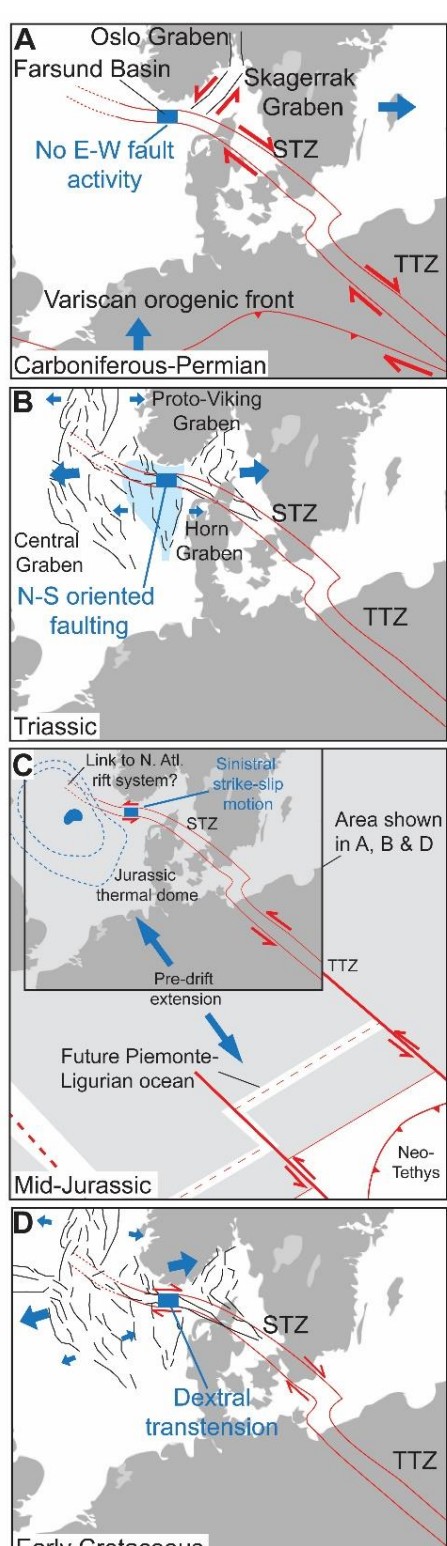

**Figure 16.** A) Schematic model showing the regional tectonic setting during the Carboniferous-Permian and linking this to the lack of observed activity within the Farsund Basin. B) Schematic model linking the regional Triassic E-W oriented extension to the Triassic N-S oriented faulting within the Farsund Basin. C) Wider-scale schematic model to highlight potential causative mechanisms for Early-Mid Jurassic sinistral strike-slip activity in the Farsund Basin. Viable driving mechanisms include the uplift of the mid-north sea thermal dome and the opening of the Piemont-Ligurian Ocean. D) Schematic model linking the local Early Cretaceous dextral transtension observed within the Farsund Basin to regional E-W to NE-SE extension occurring at this time.