# Peer review of "Oblique reactivation of lithosphere-scale lineaments controls rift physiography – The upper-crustal expression of the Sorgenfrei-Tornquist Zone, offshore southern Norway"

_Solid Earth, 2017_

## Referee Comment (RC1) · Anonymous Referee #1 · 2 Oct 2017

Review of the paper Title: Oblique reactivation of lithosphere-scale lineaments controls rift physiography - The upper crustal expression of the Sorgenfrei-Tornquist Zone offshore southern Norway Author(s): Thomas B. Phillips et al. MS No.: se-2017-97 MS Type: Research article

In this paper, the Authors present a detailed analysis of the development and evolution of the Farsund Basin, offshore southern Norway, an E-trending basin believed to represent the upper crustal expression of a major lithosphere-scale lineament, the Sorgenfrei-Tornquist Zone. The analysis is based on borehole-constrained 2D and

3D seismic reflection data; it documents complex activations of faults, which reflect a multistage tectonic evolution in turn controlled by interactions between the variable regional/local stress field and the long-lived pre-existing lithospheric fabric. The results of this analysis offer insights into the role of inherited lithosphere-scale structures on the architecture of deformation at upper crustal levels.

Overall the paper is very interesting; the analysis of the dataset is very detailed and the results support the interpretations, with the complex evolution of the basin controlled by the presence of the inherited lithosphere-scale structure and the variable stress field. The only doubt I have is with the idea of this basin representing a pull-apart (at least a 'classic' pull-apart). Indeed, many of the structures typically associated with these strike-slip basins do not seem to be present in the Farsund Basin (at least in the investigated area). For instance, basin sidewall faults or cross-basins faults seem to be lacking; similarly, offset segments of the major strike-slip faults (principal deformation zones, see Dooley and Schreurs 2012) are not very clear (for instance from Fig. 15). Deformation as illustrated in Fig 15 (or even in the more regional sketch of Fig 16) seems to be more similar to a 'distributed transtension' (Dooley and Schreurs 2012) than to a typical pull-apart. Anyway, I think the Authors should address this in more detail throughout the manuscript.

The stress field indicated in Fig. 15B (Early Jurassic) as also portrayed in Fig. 16 should involve some extensional displacement along the roughly E-W faults. Also, in Fig 15C there is no widening of the basin associated to the dextral transtension (i.e. it could be better to increase the distance between the two systems of faults bounding the basin to the North and South passing from panel B to panel C)

Other technical corrections 115. The 'Tornquist fan' does not seem to be indicated in Figure 1a 220. Cheng et al 2017 - reference not needed here 385. last lines not clear to me 740. n/a-n/a - please check this Fig. 1 Panel A. The rectangle indicating the location of Fig1c seems to be too large.

---

## Short Comment (SC1) · 15 Dec 2017

This paper presents a thorough analysis of seismic reflection data offshore southern Norway in order to investigate the role of pre-existing structures in the development of the region. Overall, I found the paper to be very well written, organised and insightful. The methods used are suitable for this investigation and the conclusions appear to be supported by the results. The figures depicting seismic lines are well presented, particularly when both the interpreted and uninterpreted sections are shown. The conclusion that a new phase of deformation across the North Sea occurred is arguably this con-

tributions most significant finding. I would therefore like to recommend publication in Solid Earth if the relatively minor points suggested here are considered. These minor points should not be too onerous on the authors, but I believe that they will improve the manuscript, and in particular the legibility of the figures.

First, the fault profiles are very informative and should be commended. However, they have been constructed for throw rather than offset (or heave), and thus do not account for any horizontal displacement. This seems both reasonable and inevitable, given the nature of the data. However, if there are any caveats associated with this approach then they should be stated or discussed in the manuscript potentially by expanding section 3.2. For example would the same conclusions have be drawn from analysis of fault heave, rather than throw? Furthermore, given that some spatial variation in velocity will be inevitable within the basin and that throw is measured in time, rather than depth, is a throw measured in time on one section of a fault comparable to a throw measured in time elsewhere on the fault (which could be > 40 km away)? Essentially, it would be beneficial to add a few lines to the methodology clarifying why the approach is reasonable.

My final points relate to the figures, which on the whole compliment the text very well but could undergo some minor amendments that would significantly improve the overall quality of the manuscript. First, the text on most of the figures is very small. For example the labels on Figs. 1 and 10, in addition to all the annotations on the interpreted seismic lines will be difficult to read at publication size. On Figs. 2, 4, 5, 9, 11 and 14, the insert of the location map that includes the seismic line location is too small and the white line is difficult to see against the grey background. Also a colour bar is missing from Fig. 13 and the colour bars on Fig. 6 are too small to read. A horizontal scale is missing from Fig. 2 and it would also be helpful to include the approximate location of the schematic cross section shown in Fig. 1D on one of the location maps.

---

## Referee Comment (RC2) · P. Cadenas (Referee) · 2 Jan 2018

The manuscript presents a detailed analysis of the geometry and the kinematic evolution of major structures controlling the geometry of the Farsund Basin, offshore southern Norway. The study relies on the interpretation of borehole-constrained 3D and 2D seismic reflection data, the development of isochron and thickness maps of key stratigraphic horizons, and the development of throw-length and backstripped profiles for the major interpreted structures. The authors recognize major N-S and E-W upper crustal fault populations that they relate in depth to the Sorgenfrei-Tornquist lithospheric

lineation. From the tectono-stratigraphic and the structural analysis, the authors document a polyphase activity of these faults which were reactivated in a broad range of tectonic styles during succesive stages governed by a distinctive stress field. Based on all these observations, the authors discuss the evolution and the role of long-lived pre-existing structures during subsequent rift events and debate the main geodynamic implications into the regional tectonic framework. The study is well-supported by the available data and the methods and the workflow used are appropriate for this investigation. Overall, the manuscript is well writen and follows a clear layout; the title reflects the content of the paper and the abstract provides a complete summary of the work. The text explanations are well supported by the figures, which are of a high quality. I find this study very interesting and insightful due not only to the scientific backgroung but also because of the methodology; the work drove to major conclusions that allows advances in the understanding of the studied area. The multiphase activity and the structural link between different fault systems in the Farsund Basin and the discovering of a previously undocumented Early Jurassic period of sinistral strike-slip activity are the most significant findings. In a more general sense, this work provides new insights to comprehend the constraints imposed by inherited lithospheric structures on the development of upper crustal faults during subsequent rift events, depending on the regional stress field. I would therefore highly recommend this manuscript to be published at Solid Earth. Hovever, I find that including some additional information and some minor modifications, in addition to other reviewers and readers comments, can improve the readability and the high quality of the manuscript. âǍć In the abstract, I would propose presenting the intepreted link between the upper crustal faults and the STZ after addressing the detailed analysis of the upper crustal faults, which is the basis of the study. Thus, I would move information from lines 17-18 to line 28, just before denoting the inferred evolution of the STZ from the analysis of upper crustal faults. âǍć Line 89, figure 1a and c? and Line 90, figure 1a and c (Varnes Graben not in figure 1c). âǍć I would put the regional geological history as section 2.1. I think it could be helpful to contextualize the evolution of the STZ. âǍć I would define a section: 2.2) The

Farsund Basin and I would move the STZ to a section 2.3. I think that a section dealing with the structure and the tectonic evolution of the Farsund Basin would be neccesary to set up the context of this work. I would move lines 164 to 169 to this section and I would include in this 2.2 section an explanation about the geometry of the main faults controlling the architecture of the Farsund Basin (i.e., including the information now in lines 212 to 215, line 220; if these structures were defined by previous works, what is known about these structures from these works?, what is the length of these structures?, explained now in sub-section 4.2, supported by the detailed structural map provided in figure 1c, in case this map was developed from the intergration of previous maps). I think this could be helpful to follow the detailed analysis of the upper crustal fault population provided later by the authors. • Line 98, the Carpathian orogenic front and the Ronne Graben are not labelled in figure 1a. • Lines 108 and 109, Palaeozoic terranes belonging to Central and Western Europe, figure 1D. This cannot clearly be inferred from the figure. • Line 114, Tornquist Fan is not labelled in the referenced Figure 1A. • I would move line 119 to 122 to line 116, after explaining that the STZ is defined as a change in litospheric thickness. Then, I would explain the expression of this structure at shallow crustal levels. • Line 125, "the STZ acted?", line 165, Central Graben is not labelled in Figure 1A. • Lines 170 to 173, this information could be more suitable for a discussion. • In the data section, I think it would be necessary indicating some information about the boreholes and the seismic data (date of acquisition, adquired by oil companies, adquisition and processing parameters,. . .). Or providing a reference in case this information has been provided in another publication. • I find useful adding some references to supoport the description of the quality of the seismic records (e.g. line 184). • I would define a section 3.2) Methodoly, including the information from line 185 to line 205. I would remove the section quantitative fault analysis (if this section is preserved, I would define a previous section in the same level, explaining the seismo-stratigraphic and structural seismic interpretation, supported by the figures displaying interpreted seismic profiles, borehole analysis and isochron and thickness maps; this information is now explaining in the section dealing

with the available dataset). I would be convenient to introduce in this section what figures support each method (e.g., figure 3 after (. . .) isochrons between them (. . .) in line 190, figure 7 and 8 for the throw-length and backstripping techniques,...). • Do the intrinsic geometric uncertainties in time domain and the spatial variations of velocity values affect throw measurements? It would be convenient adding in the methodology section some lines to further explain this limitations. • Lines 207 to 210, section 4.1 should be explained before section 4.2. • I would suggest the definition of a section 4.1 dealing with the stratigraphic architecture and the supported by the figures showing the interpreted seismic profiles. I would explain in this section the main interpreted key horizons, the main units, and its seismic expression (the paragraph included now lines 237 to 240, paragraph between lines 271 and 274). I think that some seismic to well ties should be displayed in the figures to support time constraints of the major seismic horizons for the entire seismo-stratigraphic sequence. In the same way, the authors provide in most cases depth estimations for some horizons, faults displacements and surfaces, so any seismic to well ties and velocity models developed from the check-shots should be displayed to support this information. For instance, Figure 1B is of a very good quality and it has not been referred and explained within the text. I think this figure should not be part of the "tectonic setting" figures. For sure, seismic to well ties and seismo-stratigraphic analysis has been a really important part of the analysis. This is a time-consuming and toughful work but of a great interest to support the analysis of the tectonic evolution. The crustal-scale faulting analysis provided by the authors now as section 4.1 can be included in this section. I think it could be suitable providing a description about the seismic imaging and interpretation of the main faults (if the faults are introduced in the tectonic setting, the authors can describe straighforward these faults using the seismic interpretation (depth, dip, fault links. . . as it has been interpreted on the seismic profiles). I would suggest to keep the regional description based on isochron maps in section 4.2. • Line 215, Varnes Graben is not labelled in Figure 1c. • Line 216, the Fjerritslew North and South merge with the Farsund Fault between 6 and 8 s (TWT)?. • Line 225-228, the Moho-related reflectivity across the

Fjerritslev Fault system can be inferred in figure S2 provided as supplementary material. Where is this profile located?. Has this feature been observed on several seismic profiles and/or previoysly proposed by noted references? (if this is the case this should be explained in the setting and overcome in the discussion to debate the link between the upper crustal faults and the STZ as deduced from this study and what has been previously proposed). Do the link between the STZ and the upper crustal faults is deduced from the lack of Moho reflections beneath the Farsund Basin, from the offset of Moho reflections by the Fjerritslev Fault System and from the distinctive trend of the upper crustal faults within the Farsund Basin when compared with the trend of the structure delineating the North Sea rift? • Line 256-257, the authors suggest that the Fjerritslev North and South faults merge south of the 3D dataset as indicated by 2D seismic data. A figure displaying a 2D seismic line should be provided as a reference to support this interpretation?. • Line 260, HF2 is not labelled in the isochron maps showing the structure of the supra-salt levels. • Line 304, the NS1 and NS2 faults are not labelled in figure 6b. • Line 322, although slip=fault activity?? • Lines 332 to 334, for the methodology section? After explaining the main developed isochron maps and before explaining the throw-length and backstripping profiles?. • I would suggest including a discussion section dealing with polyphase fault activity. I find the kinematic evolution of these faults described in section 6 from direct observations very suitable. However, some other aspects as the discrete Triassic activity along some segments of E-W faults (included between lines 368 and 389) or the geometric evolution of these faults during the Late Jurassic-Early Cretaceous oblique reactivation (lines 431 to 440, 445 to 447, 452 to 457 and 478 to 484) are discussed together with the previous interpretations and theoretical concepts so I find that these parts could be more appropriate for a discussion section. • Line 397, Figure 10c? • Line 482, the eastern part of the Fjerritslev South Fault? • Line 625-630, do the authors mean that the STZ accomodated most of the deformation allowing the preservation of the cratonic lithosphere of Baltica almost undeformed?.

Figures Figure 1 • It would be useful including the meaning of STZ, TTZ and RFH in

the figure caption of figure 1A. The rectangle delineating the extend of figure 1C seems to be too large taking into account the intersection of Fjerritslev Faults. • Figure 1B is not cited and explained in the text. • Figure 1C does not have a north arrow; It would be useful including boreholes in Figure 1C together with the 2D seismic profiles. What is the red dot in figure 1c? a well location?. • The fault network across the Farsund Basin showed in Figure 1c. Taking into account some references provided in the text, I suppose that it comes from previous studies, but was this map developed from previous studies or was this map developed relying on seismic interpretions made in this study? If the fault trends come from previous studies, these references should be added in the figure caption. If the map was developed during this study it is a new outcome and it should be included in other figure to support the analysis of upper crustal faults. Figures 2,4,5,9,10 and 14 • Salt diapirs should be labelled or included in the legend. • 2D seismic lines: this should be included in the figure caption. • The inset map should include the scale; the maps are too small and the seismic lines displayed in white are difficult to see against the grey background. Figure 3, 6 and 12 • Isochron maps showing thickness variations and associated faults for the main interpreted horizons; A) Triassic; B) Jurassic; C) Lower Cretaceous. • It could be useful adding latitudes and longitudes labels and using them as a reference to describe some of the observed features within the text. • It should also useful adding some key contour values. Figure 15 • It could be useful adding fault names.

---

## Author Comment (AC1) · 5 Mar 2018

Oblique reactivation of lithosphere-scale lineaments controls rift physiography – The upper crustal expression of the Sorgenfrei-Tornquist Zone, offshore southern Norway Thomas B. Phillips; Christopher A-L. Jackson; Rebecca E. Bell; Oliver B. Duffy Solid Earth MS No.: se-2017-97 Reviewer 1 – Anonymous referee We thank the referee for their overall positive review and for their constructive comments that will undoubtedly improve the manuscript. We now detail our responses to the individual comments raised in the review (highlighted in italics). These changes have been included in the

attached tracked-changes document.

Original review

In this paper, the Authors present a detailed analysis of the development and evolution of the Farsund Basin, offshore southern Norway, an E-trending basin believed to represent the upper crustal expression of a major lithosphere-scale lineament, the Sorgenfrei-Tornquist Zone. The analysis is based on borehole-constrained 2D and 3D seismic reflection data; it documents complex activations of faults, which reflect a multistage tectonic evolution in turn controlled by interactions between the variable regional/local stress field and the long-lived pre-existing lithospheric fabric. The results of this analysis offer insights into the role of inherited lithosphere-scale structures on the architecture of deformation at upper crustal levels. Overall the paper is very interesting; the analysis of the dataset is very detailed and the results support the interpretations, with the complex evolution of the basin controlled by the presence of the inherited lithosphere-scale structure and the variable stress field. The only doubt I have is with the idea of this basin representing a pull-apart (at least a 'classic' pull-apart). Indeed, many of the structures typically associated with these strike-slip basins do not seem to be present in the Farsund Basin (at least in the investigated area). For instance, basin sidewall faults or cross-basins faults seem to be lacking; similarly, offset segments of the major strike-slip faults (principal deformation zones, see Dooley and Schreurs 2012) are not very clear (for instance from Fig. 15). Deformation as illustrated in Fig 15 (or even in the more regional sketch of Fig 16) seems to be more similar to a 'distributed transtension' (Dooley and Schreurs 2012) than to a typical pull-apart. Anyway, I think the Authors should address this in more detail throughout the manuscript. Author response - We agree that the main characteristic features of a pull-apart basin (e.g. clear principal displacement zones and basin sidewall faults) cannot be determined in this area. As such, we have removed the reference to a pull-apart basin and also to transtension (Line 589-593; 575-578). Instead, we now use the term "oblique reactivation of the fault", and expand upon our previous points by stating that

the dominant motion is extensional, but with an oblique component of displacement (Line 587-588) (see also, changes to the arrows depicting the regional stress field in Figure 15).

The stress field indicated in Fig. 15B (Early Jurassic) as also portrayed in Fig. 16 should involve some extensional displacement along the roughly E-W faults.

Authors response - Although there may be some extensional displacement along the faults during the Early Jurassic, we are unable to identify this due to erosion at the Base Jurassic Unconformity. However, we have now adapted the arrows in Figure 15 to indicate the wide range of potential stress fields.

Also, in Fig 15C there is no widening of the basin associated to the dextral transtension (i.e. it could be better to increase the distance between the two systems of faults bounding the basin to the North and South passing from panel B to panel C)

Authors response - We have incorporated this change into the figure.

Other technical corrections 115. The 'Tornquist fan' does not seem to be indicated in Figure 1a

Authors response - The location of the Tornquist Fan has been noted in the text and reference to the figure has been removed (Line 164). Due to the super-regional nature of this structure, we feel that its exact location is not relevant to this study

220. Cheng et al 2017 - reference not needed here

Authors response - Reference to Cheng et al. (2017) has been removed. (Line 262)

385. last lines not clear to me

Authors response - This section of the text has now been made clearer (Line 436-438)

740. n/a-n/a - please check this

Authors response - References have been updated accordingly throughout

Fig. 1 Panel A. The rectangle indicating the location of Fig1c seems to be too large.

Authors response - Agreed, box size has been corrected
* * *

---

## Author Comment (AC2) · 5 Mar 2018

Oblique reactivation of lithosphere-scale lineaments controls rift physiography – The upper crustal expression of the Sorgenfrei-Tornquist Zone, offshore southern Norway Thomas B. Phillips; Christopher A-L. Jackson; Rebecca E. Bell; Oliver B. Duffy Solid Earth

MS No.: se-2017-97

Reviewer 2 – Dr. Patricia Cadenas

[Figure]

We thank Dr Cadenas for her insightful and thorough review, which we believe will greatly improve the overall manuscript. Following the overall positive nature of the review (see below), we append detailed responses to each of the queries and comments raised. Changes to the overall manuscript can be found in the attached tracked-changes document.

Original review

The manuscript presents a detailed analysis of the geometry and the kinematic evolution of major structures controlling the geometry of the Farsund Basin, offshore southern Norway. The study relies on the interpretation of borehole-constrained 3D and 2D seismic reflection data, the development of isochron and thickness maps of key stratigraphic horizons, and the development of throw-length and backstripped profiles for the major interpreted structures. The authors recognize major N-S and E-W upper crustal fault populations that they relate in depth to the Sorgenfrei-Tornquist lithospheric lineation. From the tectono-stratigraphic and the structural analysis, the authors document a polyphase activity of these faults which were reactivated in a broad range of tectonic styles during successive stages governed by a distinctive stress field. Based on all these observations, the authors discuss the evolution and the role of long-lived pre-existing structures during subsequent rift events and debate the main geodynamic implications into the regional tectonic framework. The study is well-supported by the available data and the methods and the workflow used are appropriate for this investigation. Overall, the manuscript is well written and follows a clear layout; the title reflects the content of the paper and the abstract provides a complete summary of the work. The text explanations are well supported by the figures, which are of a high quality. I find this study very interesting and insightful due not only to the scientific background but also because of the methodology; the work drove to major conclusions that allows advances in the understanding of the studied area. The multiphase activity and the structural link between different fault systems in the Farsund Basin and the discovering of a previously undocumented Early Jurassic period of sinistral strike-slip activity

are the most significant findings. In a more general sense, this work provides new insights to comprehend the constraints imposed by inherited lithospheric structures on the development of upper crustal faults during subsequent rift events, depending on the regional stress field. I would therefore highly recommend this manuscript to be published at Solid Earth. However, I find that including some additional information and some minor modifications, in addition to other reviewers and readers comments, can improve the readability and the high quality of the manuscript. In the abstract, I would propose presenting the interpreted link between the upper crustal faults and the STZ after addressing the detailed analysis of the upper crustal faults, which is the basis of the study. Thus, I would move information from lines 17-18 to line 28, just before denoting the inferred evolution of the STZ from the analysis of upper crustal faults.

Authors response - We agree with this comment from the reviewer. We have changed the structure of the abstract so that the analysis of the upper crustal fault populations is first introduced and then linked to the lithosphere-scale STZ (Line 17-34).

Line 89, figure 1a and c? and Line 90, figure 1a and c (Varnes Graben not in figure 1c).

Authors response - We have made the required changes to the references to this figure. A label for the Varnes Graben has been included into Figure 1c.

I would put the regional geological history as section 2.1. I think it could be helpful to contextualize the evolution of the STZ.

Authors response - We agree with this comment and as such now establish the framework of the regional geological history, before placing the STZ into this context. (Line 100-141)

I would define a section: 2.2) The Farsund Basin and I would move the STZ to a section 2.3. I think that a section dealing with the structure and the tectonic evolution of the Farsund Basin would be neccessary to set up the context of this work. I would move

lines 164 to 169 to this section and I would include in this 2.2 section an explanation about the geometry of the main faults controlling the architecture of the Farsund Basin (i.e., including the information now in lines 212 to 215, line 220; if these structures were defined by previous works, what is known about these structures from these works?, what is the length of these structures?, explained now in sub-section 4.2, supported by the detailed structural map provided in figure 1c, in case this map was developed from the integration of previous maps). I think this could be helpful to follow the detailed analysis of the upper crustal fault population provided later by the authors.

Authors response - Having carefully considered this suggestion, we disagree that the Farsund Basin should have its own sub-section in the Geological background section of the paper. Although the approximate geometry and existence of the basin has been defined previously by other authors, the detailed geometry and kinematics of the faults are new to this study (ie the individual faults have never been described in detail before or their properties including length characterised- this is new to our study). We therefore feel detailed mapping of the geometry of the basin forms a fundamental part of the new results presented in this study (and this information should be presented in the Results section rather than Geological Background), and are vital in linking the basin to the deeper lineament.

Line 98, the Carpathian orogenic front and the Ronne Graben are not labelled in figure 1a.

Authors response - These super-regional structures are currently not shown on this figure, as we mainly focus on the northwestern component of the Tornquist zone (i.e. the Sorgenfrei-Tornquist Zone). References to more regional studies have been included to highlight the more regional structures (Line 146-147) (although we do not feel these super-regional structures are directly relevant to our study).

Lines 108 and 109, Palaeozoic terranes belonging to Central and Western Europe, figure 1D. This cannot clearly be inferred from the figure.

Authors response - This sentence has now been clarified, and the figure reference removed (Line 158).

Line 114, Tornquist Fan is not labelled in the referenced Figure 1A.

Authors response - We have removed the figure reference from this section (Line 164). The location of this more regional structure is not of importance to this study, although some information is added to the caption of Figure 1

I would move line 119 to 122 to line 116, after explaining that the STZ is defined as a change in lithospheric thickness. Then, I would explain the expression of this structure at shallow crustal levels.

Authors response - We agree with the reviewer and have reorganised this section such that the evidence showing that at upper-crustal levels the STZ resides within Baltica (Line 166-169) comes before the definition of the STZ at these upper crustal levels (Line 170).

Line 125, "the STZ acted?",

Authors response - The phrase "acts as" has been changed to "represents" (Line 178)

Line 165, Central Graben is not labelled in Figure 1A.

Authors response - Reference to the Central Graben has been removed (Line 133)

Lines 170 to 173, this information could be more suitable for a discussion.

Authors response - We feel that this section provides evidence for the basin being linked to a pre-existing structure, a key platform which later interpretations are based on. Therefore, we believe this information should come at this point in the paper.

In the data section, I think it would be necessary indicating some information about the boreholes and the seismic data (date of acquisition, acquired by oil companies, acquisition and processing parameters,. . .). Or providing a reference in case this

information has been provided in another publication.

Authors response - Additional information regarding the processing and acquisition information of the seismic surveys used in this study has been made available in a table in the supplementary material (Line 191-192).

I find useful adding some references to support the description of the quality of the seismic records (e.g. line 184).

Authors response - Figure references have since been added to this section to visually support these textual descriptions (Line 196)

I would define a section 3.2) Methodology, including the information from line 185 to line 205. I would remove the section quantitative fault analysis (if this section is preserved, I would define a previous section in the same level, explaining the seismo-stratigraphic and structural seismic interpretation, supported by the figures displaying interpreted seismic profiles, borehole analysis and isochron and thickness maps; this information is now explaining in the section dealing with the available dataset). I would be convenient to introduce in this section what figures support each method (e.g., figure 3 after (. . .) isochrons between them (. . .) in line 190, figure 7 and 8 for the throw-length and backstripping techniques,...).

Authors response - We have added an additional section (now section 3.2) that details the seismic interpretation and the generation of the isochrons (Line 205-298). Figure references have been changed where appropriate, but we have ensured that the figures are introduced in the order they appear are in the text.

Do the intrinsic geometric uncertainties in time domain and the spatial variations of velocity values affect throw measurements? It would be convenient adding in the methodology section some lines to further explain this limitations.

Authors response - Because the overlying sedimentary sections are relatively homogeneous along-strike of the fault we expect there to be little lateral variation in overburden

velocity (and thus extracted fault throws in the deeper section), we argue that, although absolute values (in metres) may vary, the overall throw patterns (in two-way time) underpinning our key interpretation and conclusions will remain. In addition, the faults are roughly at the same depth from east to west, thus burial related changes in velocity are also not expected to greatly effect depth conversions. To further support our interpretations, we undertook spot depth conversions (Line 200-201), with the appropriate depth measurements cited in the text (e.g. Line 268, Line 351, Line 499).

Lines 207 to 210, section 4.1 should be explained before section 4.2.

Authors response - Sentence order in this section has now been changed (Line 244-248).

I would suggest the definition of a section 4.1 dealing with the stratigraphic architecture and the supported by the figures showing the interpreted seismic profiles. I would explain in this section the main interpreted key horizons, the main units, and its seismic expression (the paragraph included now lines 237 to 240, paragraph between lines 271 and 274). I think that some seismic to well ties should be displayed in the figures to support time constraints of the major seismic horizons for the entire seismo-stratigraphic sequence. In the same way, the authors provide in most cases depth estimations for some horizons, faults displacements and surfaces, so any seismic to well ties and velocity models developed from the checkshots should be displayed to support this information. For instance, Figure 1B is of a very good quality and it has not been referred and explained within the text. I think this figure should not be part of the "tectonic setting" figures. For sure, seismic to well ties and seismo-stratigraphic analysis has been a really important part of the analysis. This is a time-consuming and toughful work but of a great interest to support the analysis of the tectonic evolution. The crustal-scale faulting analysis provided by the authors now as section 4.1 can be included in this section. I think it could be suitable providing a description about the seismic imaging and interpretation of the main faults (if the faults are introduced in the tectonic setting, the authors can describe straighforward these faults using the seismic interpretation

(depth, dip, fault links. . . as it has been interpreted on the seismic profiles). I would suggest to keep the regional description based on isochron maps in section 4.2.

Authors response - We cannot generate seismic-well ties due to a lack of well-log data in the key wells. Checkshot data from multiple wells within and outside of the immediate area of interest are used for the depth conversion; this is now explained in the methodology section (Line 199-202). As mentioned in response to a previous comment, we feel that the detailed description of the faults should not be included in the regional setting section as this is new data generated in this study.

Line 215, Varnes Graben is not labelled in Figure 1c.

Authors response - Label has been added

Line 216, the Fjerritslev North and South merge with the Farsund Fault between 6 and 8 s (TWT)?

Authors response - Sentence has been clarified. (Line 259-260)

Line 225-228, the Moho-related reflectivity across the Fjerritslev Fault system can be inferred in Figure S2 provided as supplementary material. Where is this profile located?. Has this feature been observed on several seismic profiles and/or previously proposed by noted references? (if this is the case this should be explained in the setting and overcome in the discussion to debate the link between the upper crustal faults and the STZ as deduced from this study and what has been previously proposed). Do the link between the STZ and the upper crustal faults is deduced from the lack of Moho reflections beneath the Farsund Basin, from the offset of Moho reflections by the Fjerritslev Fault System and from the distinctive trend of the upper crustal faults within the Farsund Basin when compared with the trend of the structure delineating the North Sea rift?

Authors response - Location has been provided for Figure S2. Less emphasis will be placed on the possible linkage shown in Figure 2, as the interpretation at depth on

this seismic section is highly uncertain. Instead, we now highlight the evidence for a geometric link between sub-crustal and upper crustal components, comparing this to proposals in different areas from other papers (i.e. Deeks and Thomas 1995) (Line 271-282).

Line 256-257, the authors suggest that the Fjerritslev North and South faults merge south of the 3D dataset as indicated by 2D seismic data. A figure displaying a 2D seismic line should be provided as a reference to support this interpretation?

Authors response - Unfortunately, due to a lack of 3D seismic reflection data and suitably orientated 2D seismic profiles imaging faults, we can only currently speculate on this relationship. This is now made clear in the manuscript (Line 306-307). We suggest a relationship based on regional mapping and 2D seismic sections located further south of the 3D seismic volume, where a single fault is present. Furthermore, the phrasing of this sentence has been changed. Based on their map-view geometry, we infer the N-S striking faults merge southwards; however, this cannot be proven due to a lack of 3D seismic data and only few 2D lines in this area.

Line 260, HF2 is not labelled in the isochron maps showing the structure of the supra-salt levels.

Authors response - HF2 has since been labelled on Figure 6

Line 304, the NS1 and NS2 faults are not labelled in figure 6b.

Authors response - Labels have been added to the figure.

Line 322, although slip=fault activity??

Authors response - "Slip" has been changed to "fault activity" (Line 372).

Lines 332 to 334, for the methodology section? After explaining the main developed isochron maps and before explaining the throw-length and backstripping profiles?.

Authors response - This information has now been incorporated into the methodology

section (Line 213-215).

I would suggest including a discussion section dealing with polyphase fault activity. I find the kinematic evolution of these faults described in section 6 from direct observations very suitable. However, some other aspects as the discrete Triassic activity along some segments of E-W faults (included between lines 368 and 389) or the geometric evolution of these faults during the Late Jurassic-Early Cretaceous oblique reactivation (lines 431 to 440, 445 to 447, 452 to 457 and 478 to 484) are discussed together with the previous interpretations and theoretical concepts so I find that these parts could be more appropriate for a discussion section.

Authors response - Additional information has been incorporated into the first discussion section related to the polyphase fault activity, including the correlation between pre-existing faults and cross-cutting relay ramps, where pre-existing faults may segment the later-formed faults (Line 579-584). However, we believe that some of the information referred to in the reviewers comment (e.g. "Line 431-440 etc." (now at 483-486)) is of fundamental importance to interpretations, and therefore do not belong in the discussion section.

Line 397, Figure 10c?

Authors response - This reference has now been changed to Fig. 10c and 11

Line 482, the eastern part of the Fjerritslev South Fault?

Authors response - No, this refers to the western segment of the fault. Information regarding the segmentation of the fault has been added to Figure 12.

Line 625-630, do the authors mean that the STZ accommodated most of the deformation allowing the preservation of the cratonic lithosphere of Baltica almost undeformed?

Authors response - Yes, this is what was meant. This sentence has been altered to make this point clearer (Line 689-690).

Figure 1 It would be useful including the meaning of STZ, TTZ and RFH in the figure caption of figure 1A.

Authors response - The definitions of these abbreviations have been included in the Figure caption.

The rectangle delineating the extend of figure 1C seems to be too large taking into account the intersection of Fjerritslev Faults.

Authors response - This has been changed.

Figure 1B is not cited and explained in the text.

Authors response - This figure has now been cited in the seismic interpretation section of the methodology (Line 207-209

Figure 1C does not have a north arrow;

Authors response - Changed

It would be useful including boreholes in Figure 1C together with the 2D seismic profiles. What is the red dot in figure 1c? a well location?.

Authors response - The only available borehole in the area is currently shown in Figure 1c by the red dot. The definition of the red dot is further explained in the figure caption along with the figure inset.

The fault network across the Farsund Basin showed in Figure 1c. Taking into account some references provided in the text, I suppose that it comes from previous studies, but was this map developed from previous studies or was this map developed relying on seismic interpretions made in this study? If the fault trends come from previous studies, these references should be added in the figure caption. If the map was developed during this study it is a new outcome and it should be included in other figure to support the analysis of upper crustal faults.

Authors response - This fault map has been created through interpretations made in this study, rather than compiled from previous works. Therefore, this forms a major result of this study, the origin of this fault map is now stated in the caption. Regional faults are based on the NPD fault maps; this is now stated in the figure caption.

Figures 2,4,5,9,10 and 14 Salt diapirs should be labelled or included in the legend.

Authors response - Salt has been added to the captions or the figures of the seismic sections where appropriate

2D seismic lines: this should be included in the figure caption. Authors response - Changed

The inset map should include the scale; the maps are too small and the seismic lines displayed in white are difficult to see against the grey background.

Authors response - Rough scale is included in the inset maps. Colour of the seismic lines has been changed to make them stand out more. Figure 3, 6 and 12 Isochron maps showing thickness variations and associated faults for the main interpreted horizons; A) Triassic; B) Jurassic; C) Lower Cretaceous.

It could be useful adding latitudes and longitudes labels and using them as a reference to describe some of the observed features within the text.

Authors response - The main features have been identified based on fault segments currently. The labelling of these different fault segments has been made clearer on the maps.

It should also useful adding some key contour values.

Authors response - Contour interval has been included in the caption.

Figure 15 It could be useful adding fault names.

Authors response - Fault names have been added to Figure 15c, as this panel
represents the present day fault network.

Please also note the supplement to this comment:
https://www.solid-earth-discuss.net/se-2017-97/se-2017-97-AC2-supplement.pdf

—————————————————————

---

## Author Comment (AC3) · 5 Mar 2018

Oblique reactivation of lithosphere-scale lineaments controls rift physiography – The upper crustal expression of the Sorgenfrei-Tornquist Zone, offshore southern Norway Thomas B. Phillips; Christopher A-L. Jackson; Rebecca E. Bell; Oliver B. Duffy

Solid Earth

MS No.: se-2017-97

[Figure]

Short comment 1 – Dr. Alexander Peace

Original comment

This paper presents a thorough analysis of seismic reflection data offshore southern Norway in order to investigate the role of pre-existing structures in the development of the region. Overall, I found the paper to be very well written, organised and insightful. The methods used are suitable for this investigation and the conclusions appear to be supported by the results. The figures depicting seismic lines are well presented, particularly when both the interpreted and uninterpreted sections are shown. The conclusion that a new phase of deformation across the North Sea occurred is arguably this contributions most significant finding. I would therefore like to recommend publication in Solid Earth if the relatively minor points suggested here are considered. These minor points should not be too onerous on the authors, but I believe that they will improve the manuscript, and in particular the legibility of the figures. First, the fault profiles are very informative and should be commended. However, they have been constructed for throw rather than offset (or heave), and thus do not account for any horizontal displacement. This seems both reasonable and inevitable, given the nature of the data. However, if there are any caveats associated with this approach then they should be stated or discussed in the manuscript potentially by expanding section 3.2.

Authors response - The profiles shown here only measure the vertical component of deformation (i.e. throw) associated with faulting; as stated in the main body of the text we make no inferences regarding the mode of displacement from these plots (Line 237-239). Additional information on the generation and interpretation of these fault profiles is available in Appendix A and B.

For example would the same conclusions have be drawn from analysis of fault heave, rather than throw?

Authors response - The width of the fault polygons shown in the figures (i.e. Figure 3, 6) provides a first-order approximation of fault heave, a point now expanded on in the

text (Line 222-224). However, we believe that fault heave is unable to provide more information than that provided by throw, particularly regarding the fault geometry and kinematics. Furthermore, for extension, throw is typically much greater than heave; therefore, by measuring throw we are able to minimise any errors associated with our measurements, making our interpretations more robust (Line 225-227).

Furthermore, given that some spatial variation in velocity will be inevitable within the basin and that throw is measured in time, rather than depth, is a throw measured in time on one section of a fault comparable to a throw measured in time elsewhere on the fault (which could be > 40 km away)? Essentially, it would be beneficial to add a few lines to the methodology clarifying why the approach is reasonable.

Authors response - Although the absolute values of the throw may change we believe that the overall patterns and locations of major throw changes would remain the same after depth conversion. Unfortunately, there is limited well-log data or seismic velocity analyses for us to determine the velocity structure in this manuscript. Within the 3D volume the overall overburden is relatively homogeneous along the fault and therefore velocity is likely to be relatively constant along-strike of the faults. Furthermore, the faults are roughly at the same depth from east to west, thus burial related changes in velocity are also not expected to greatly effect depth conversions. In order to test the difference that depth conversion would make to our throw profiles we depth converted structural measurements made at various points along the fault using limited velocity information (checkshot data) from regional wells (Line 200-201). The overall throw patterns and our interpretations are very similar after depth conversion to those made in the time domain.

My final points relate to the figures, which on the whole compliment the text very well but could undergo some minor amendments that would significantly improve the overall quality of the manuscript. First, the text on most of the figures is very small. For example the labels on Figs. 1 and 10, in addition to all the annotations on the interpreted seismic lines will be difficult to read at publication size. On Figs. 2, 4, 5, 9, 11 and 14,

the insert of the location map that includes the seismic line location is too small and the white line is difficult to see against the grey background. Also a colour bar is missing from Fig. 13 and the colour bars on Fig. 6 are too small to read. A horizontal scale is missing from Fig. 2 and it would also be helpful to include the approximate location of the schematic cross section shown in Fig. 1D on one of the location maps.

Authors response - All annotations and labels on figures are now of a size that will be visible when published full size and adhere to the Solid Earth guidelines. The colour bar for Figure 13 is the same as in Figure 12, this has been added to the caption. The colour bars in Figure 6 have been enlarged. A horizontal scale is already included in Figure 2 (right of figure). Information regarding the approximate location and orientation of the schematic cross-section (Fig. 1D) has been added to the figure caption. In addition, this figure has been modified to more accurately reflect the nature of the lithosphere-scale nature of the lineament, and the link between upper- and sub-crustal components as emphasised in the text (Line 271-276).

Please also note the supplement to this comment:
https://www.solid-earth-discuss.net/se-2017-97/se-2017-97-AC3-supplement.pdf

---

## Author Comment (AC4) · 5 Mar 2018

Please find attached a tracked changes document of the manuscript, detailing the changes made in response to the reviewers comments. Individual replies to comments and reviewers are made in response to each comment.

Please also note the supplement to this comment:
https://www.solid-earth-discuss.net/se-2017-97/se-2017-97-AC4-supplement.pdf